# Orbital control on the timing of oceanic anoxia in the Late Cretaceous

**S. J. Batenburg[1,2], D. De Vleeschouwer[3,4], M. Sprovieri[5], F. J. Hilgen[6], A. S. Gale[7], B. S. Singer[8], C. Koeberl[9,10], R. Coccioni[11], P. Claeys[4], A. Montanari[12]**

[1] Department of Earth Sciences, University of Oxford, Oxford, United Kingdom

[2] Institut für Geowissenschaften, Goethe-Universität Frankfurt, Frankfurt am Main, Germany

[3] MARUM, Universität Bremen, Bremen, Germany

[4] Earth System Sciences, Vrije Universiteit Brussel, Brussels, Belgium

[5] IAMC-CNR Capo Granitola, Campobello di Mazara, Italy

[6] Department of Earth Sciences, Utrecht University, Utrecht, the Netherlands

[7] School of Earth and Environmental Sciences, University of Portsmouth, Portsmouth, United Kingdom

[8] Department of Geoscience, University of Wisconsin-Madison, Madison, Wisconsin, United States of America

[9] Department of Lithospheric Research, University of Vienna, Vienna, Austria

[10] Natural History Museum Vienna, Vienna, Austria

[11] Dipartimento di Scienze della Terra, della Vita e dell'Ambiente, Università degli Studi "Carlo Bo", Urbino, Italy

[12] Osservatorio Geologico di Coldigioco, 62020 Frontale di Apiro, Italy

Correspondence to: (sbatenburg@gmail.com)

**Abstract**

The oceans at the time of the Cenomanian-Turonian transition were abruptly perturbed by a period of bottom-water anoxia. This led to the brief but widespread deposition of black organic-rich shales, such as the Livello Bonarelli in the Umbria-Marche Basin (Italy). Despite

intensive studies, the origin and exact timing of this event are still debated. In this study, we assess leading hypotheses about the inception of oceanic anoxia in the Late Cretaceous greenhouse world, by providing a 6-Myr-long astronomically-tuned timescale across the Cenomanian-Turonian boundary. We procure insights in the relationship between orbital forcing and the Late Cretaceous carbon cycle by deciphering the imprint of astronomical cycles on lithologic, geophysical, and stable isotope records, obtained from the Bottaccione, Contessa and Furlo sections in the Umbria-Marche Basin. The deposition of black shales and cherts, as well as the onset of oceanic anoxia, is related to maxima in the 405-kyr cycle of eccentricity-modulated precession. Correlation to radioisotopic ages from the Western Interior (USA) provides unprecedented age control for the studied Italian successions. The most likely tuned age for the Livello Bonarelli base is 94.17 ± 0.15 Ma (tuning #1); however, a 405-kyr older age cannot be excluded (tuning #2) due to uncertainties in stratigraphic correlation, radioisotopic dating, and orbital configuration. Our cyclostratigraphic framework suggests that the exact timing of major carbon cycle perturbations during the Cretaceous may be linked to increased variability in seasonality (i.e. a 405-kyr eccentricity maximum) after the prolonged avoidance of seasonal extremes (i.e. a 2.4-Myr eccentricity minimum). Volcanism is probably the ultimate driver of oceanic anoxia, but orbital periodicities determine the exact timing of carbon cycle perturbations in the Late Cretaceous. This unites two leading hypotheses about the inception of oceanic anoxia in the Late Cretaceous greenhouse world.

## 1   Introduction

The organic rich Livello Bonarelli formed as a result of oxygen deficiency and carbonate dissolution in the oceans during the Cenomanian/Turonian (C/T) transition. During this Ocean Anoxic Event 2 (OAE2), a combination of factors caused increased productivity, incomplete decomposition of organic matter and widespread deposition of black shales. Although these sediments are intensively studied, the exact extent, cause, timing and duration of oceanic anoxia are debated (Sinton and Duncan, 1997; Mitchell et al., 2008). Contrasting causal mechanisms have been suggested, including stratification of the water column (Lanci et al., 2010) versus intensification of the hydrological cycle driving a dynamic ocean circulation (Trabucho-Alexandre et al., 2010). Studies on trace-elemental and (radiogenic) isotope compositions of Cenomanian marine successions have suggested a volcanic origin of OAE2, by delivering nutrients to the semi-enclosed proto-North Atlantic (Du Vivier et al., 2014;

Zheng et al., 2013; and references therein). Deciphering the importance of volcanic and
oceanographic processes requires tight constraints on their relative timing. Regularly
occurring black cherts and shales below the Livello Bonarelli demonstrate that oceanic
conditions in the Umbria-Marche Basin were punctuated by episodes of regional anoxia from
the mid-Cenomanian onwards. Their hierarchical stacking pattern suggests an orbital control
on the deposition of organic rich horizons (Mitchell et al., 2008; Lanci et al., 2010). Stable
carbon isotope data reveal that long-term variations in eccentricity paced the carbon cycle
(Sprovieri et al., 2013) and sea level changes (Voigt et al., 2006) of the Late Cretaceous. Here
we investigate the role of orbital forcing on climate and the carbon cycle, and, specifically, on
organic-rich sedimentation prior, during, and after OAE2.
We also explore the potential for establishing an anchored astrochronology for the C/T
interval in Europe. Recent improvements in the astronomical solution (La2011; Laskar et al.,
2011b) and in the intercalibration of radiometric and astronomical dating techniques (Kuiper
et al., 2008; Renne et al., 2013) allow the extension of the astronomical time scale into the
Cretaceous. The C/T boundary in the Western Interior (USA) has been dated at 93.90 ± 0.15
Ma by intercalibration of radio-isotopic and astrochronologic time scales (Meyers et al.,
2012b). Also, reinterpretation of proxy records spanning the C/T interval seems to resolve
discrepancies in reported durations of the OAE2 (Meyers et al., 2012a; Sageman et al., 2006).
The well-documented Italian rhythmic successions, reference sections for climatic processes
in the Tethyan realm, need to be tied in with the numerical time scale. Biostratigraphic
correlation to radioisotopically-dated ash beds in the Western Interior is complicated by the
provinciality of faunas and floras. However, $\delta^{13}$C stratigraphy provides a reliable correlation
tool (Jenkyns et al., 1994) and we present a new $^{40}$Ar/$^{39}$Ar age for the Thatcher bentonite from
the Western Interior occurring within the mid-Cenomanian $\delta^{13}$C event (MCE). This study
integrates the well-developed cyclostratigraphy from the Umbria-Marche Basin with
radioisotopic ages from the Western Interior and derives a numerical timescale for this critical
interval in Earth's history.

## 2 Materials and Methods

### 2.1 Geological setting and proxy records

Previous studies have investigated the rhythmic nature of the bedded limestones, (black) cherts and shales in sections near Gubbio (de Boer, 1982, 1983; Herbert and Fischer, 1986; Schwarzacher, 1994; Sprovieri et al., 2013) and at Furlo (Beaudouin et al., 1996; Mitchell et al., 2008; Lanci et al., 2010). In this study, we present new geophysical and stable isotope data generated from the Cenomanian interval at the Furlo quarry, and from uppermost Cenomanian and Turonian deposits in the Gola del Bottaccione (Fig. 1). In addition, stable carbon and oxygen isotope data from the Turonian of the Contessa quarry, published by Stoll and Schrag (2000) are used. The proxy records from the Bottaccione, Contessa and Furlo sections are all presented on the same height scale, using the recent height scale for the Cretaceous Umbria-Marche basin, introduced by Sprovieri et al. (2013).

In the Umbria-Marche Basin, the Livello Bonarelli separates the Cenomanian white limestones of the Scaglia Bianca from the Turonian pink limestones of the Scaglia Rossa. The strong changes in sedimentary facies necessitate the application of different proxy methods as archives for paleoclimatic variability. Colour reflectance was measured in the Cenomanian Scaglia Bianca, to capture the alternation of black cherts and shales with white limestones and light grey cherts. For the Turonian Scaglia Rossa, where colour variations are limited, magnetic susceptibility measurements reveal variations in the detrital contribution, as clastic particles are generally richer in ferromagnetic minerals. To investigate the Livello Bonarelli in high resolution, XRF data were generated, reflecting variations in detrital contribution ($SiO_2$, $Al_2O_3$, $TiO_2$) and organic matter content (Loss on ignition, LOI). High $Al_2O_3$ likely indicates a strong riverine input, in contrast to $TiO_2$, reflecting a stronger dust contribution. The $SiO_2$ can be both detrital and biogenic in origin. The LOI data reflect the weight of volatile substances lost upon heating and give a measure of the organic content.

The W4 member of the Scaglia Bianca formation at Furlo (Coccioni, 1996) consists mainly of light grey to white pelagic biomicrite alternating with light grey nodular to bedded cherts and tabular black cherts (Mitchell et al., 2008). The section was logged in detail and sampled at 3 cm spacing with an electric handheld drill. The total light reflectance (L*, in %) was measured with a Konica Minolta Spectrophotometer CM 2002 on the surface of rock powders, recording the reflected energy (RSC) at 400 to 700 nm wavelengths in 10-nm steps

(averaged over three measurements). For the Furlo section, $\delta^{18}O$ and $\delta^{13}C$ were measured with
a GasBench II device and a ThermoElectron Delta Plus XP mass spectrometer at the IAMC-
CNR in Naples. Stable isotope ratios were measured on powders of all lithologies, and
repeated with larger sample amounts when carbonate contents were insufficient.
The overlying Livello Bonarelli consists of alternating 5-50 mm layers of organic-rich black
shale (up to 26% TOC) and lighter radiolarite (Kuroda et al., 2007). The Bonarelli interval
was sampled at a 2 cm resolution at both the Furlo and Bottaccione sections. Contents of
major and selected minor elements from the Livello Bonarelli were determined with a Philips
PW2400 sequential X-ray fluorescence (XRF) spectrometer equipped with a Rh-excitation
source at the University of Vienna. Details of the analytical procedures and accuracies are
similar to those given in Reimold et al. (1994).
The overlying Turonian R1 member of the Scaglia Rossa formation consists of pink pelagic
limestones and marly limestones with nodular to laminar red to gray cherts (Montanari et al.,
1989). 38 m above the Livello Bonarelli at the Bottaccione section were logged in detail and
the section was sampled at 5 cm spacing from 10 m to 30 m above the Livello Bonarelli.
Magnetic Susceptibility (MS) was measured with a Bartington MS2B Dual Frequency
magnetometer at the Osservatorio Geologico de Coldigioco (averaged over three
measurements). Stable isotope ratios were measured with a Kiel III device coupled to a
ThermoFinnigan delta+XL mass spectrometer at the Vrije Universiteit Brussel.
**2.2   Ar/Ar dating**
A new $^{40}Ar/^{39}Ar$ age was obtained for the mid-Cenomanian event in the $\delta^{13}C$ record. Sample
91-0-03 is the same material used by Obradovich (1993); it is from an ash bed in the
*Conlinoceros gilberti* ammonite zone in the Western Interior (USA), commonly known as the
Thatcher bentonite. Laser fusion $^{40}Ar/^{39}Ar$ analyses of single sanidine crystals were
performed at the WiscAr laboratory, University of Wisconsin-Madison following methods
detailed in Sageman et al. (2014). A total of 53 crystals were dated. Eleven crystals that
yielded less than 98.5% radiogenic $^{40}Ar$ were excluded from the mean, as was one inherited
crystal that gave an apparent age greater than 101 Ma. Ages are calculated relative to 28.201
$\pm$ 0.046 Ma Fish Canyon sanidines (Kuiper et al., 2008) using the $^{40}K$ decay constants of Min
et al. (2000).

## 2.3    Time series analysis

Time series analysis was carried out using the MTM-method (Thomson, 1982) with LOWSPEC background estimation (Meyers, 2012), as implemented in the R package "astrochron" (Meyers, 2014). We used three 2-$\pi$ prolate tapers and confidence levels were calculated with the LOWESS-based (Cleveland, 1979) procedure of Ruckstuhl et al. (2001). The Continuous Wavelet Transform is used to decompose the one-dimensional time-series into their two-dimensional time–frequency representation. Band-pass filters are applied with Analyseries (Paillard et al., 1996). Sedimentation rates within the Scaglia Rossa and Scaglia Bianca Formations are estimated with the Evolutionary Average Spectral Misfit (E-ASM) method (Meyers and Sageman, 2007), using all frequencies for which the MTM harmonic F-test reports a line component that exceeds 80% probability. For the Bonarelli level, sedimentation rate is estimated with the standard ASM method. Predicted orbital periods for the late Cretaceous (93 Ma) are from Berger et al. (1992).

## 3    Results

### 3.1    Lithology and proxy data

The Cenomanian black shales and cherts in the Furlo section display a hierarchical stacking pattern, with groups of 2 to 4 organic-rich levels, spaced ~20 cm apart (Figs. 2 & 3). Black cherts and shales increase in number up-section, although they are lacking in the interval between 483 and 485 m. Between 483.5 m and the Livello Bonarelli, the spacing between beds increases. Despite this increase, the two thick cherts directly underlying the Livello Bonarelli display a similar grouping as the cherts throughout the section. The total reflectance record (L*, in %) captures this stacking pattern: grey and black cherts reflect little light and display shifts towards lower L* values, in contrast to the bright micritic Scaglia Bianca limestones with high L* values. Increased variability and negative values of $\delta^{13}C$ and $\delta^{18}O$ coincide with higher variability in reflectance and with the occurrence of organic rich layers.

XRF data from the Livello Bonarelli at Furlo display a marked variability at a 12-cm scale (Fig. 4). The $TiO_2$, and $Al_2O_3$ records display very similar behaviour, whereas $SiO_2$, and LOI data additionally show variation on a 40-cm scale. At Bottaccione, a marked variability can be observed at an 8-cm scale in the $SiO_2$, $TiO_2$ and $Al_2O_3$ data from the Livello Bonarelli (Fig. 5).

The Scaglia Rossa pelagic limestones were studied in the classic Contessa and Bottaccione sections near Gubbio. Oscillations between radiolarian cherts and foram-coccolith pelagic limestones show hierarchical bundles of 2 to 5 chert layers per bundle. These bundles could be correlated amongst the Contessa and Bottaccione sections and are indicated by brackets on Figure 3. The lithologic log shown on Figure 3 is for the Bottaccione section. The magnetic susceptibility signal of the Bottaccione section accentuates the hierarchical stacking pattern, showing an increased magnetic susceptibility signal in intervals characterized by frequent chert beds (Fig. 3).

## 3.2  Ar/Ar

The inverse variance weighted mean age of 41 of the 53 sanidine crystals measured from sample 91-0-03 give an age of 96.21 ± 0.16/0.36 Ma (2σ analytical uncertainty/full uncertainty including decay constant and standard age), with an MSWD of 0.69 (Fig. 6). The complete set of analytical and standard data is in Suppl. Table 1.

## 3.3  Time series analysis

Spectral analyses by MTM/LOWSPEC, in combination with the evolutionary Average Spectral Misfit (ASM; Meyers and Sageman, 2007) method, suggest an average sedimentation rate around 11 m/Myr throughout the studied interval, excluding the Livello Bonarelli (Figs. 7 & 8).

In the Cenomanian interval, the MTM/LOWSPEC spectra of all proxies exhibit a spectral peak exceeding the 95% CL for a cycle thickness of 0.25 m, corresponding to the spacing between individual chert layers, which is interpreted as the imprint of ~21-kyr precession (left column in Fig. 7). The average accumulation rate is 11 m/Myr (lower panel in Fig. 8) and the dominant periodicities at 1 and 4 m correspond to 100-kyr and 405-kyr eccentricity, respectively (Fig. 7).

Similarly, in the Turonian interval, a 0.25 m spectral peak exceeds the 95% CL for all proxies and is interpreted as the imprint of precession (right column in Fig. 7). Here, the eccentricity components are represented by dominant periodicities of 4.66 and 1.16 m and the average accumulation rate is 10.5 m/Myr (upper panel in Fig. 8).

We also find a statistically significant imprint of obliquity in Furlo's Cenomanian $\delta^{13}$C record which confirms an important obliquity-control on the greenhouse carbon cycle, as suggested

by Laurin et al. (2015). Grouping of precession-related chert-limestone alternations in ~100-kyr bundles is indicated by brackets next to the lithological log in Fig. 3 and 405-kyr eccentricity cycles are denoted by yellow-white alternating bands. The definition of eccentricity minima and maxima is based on the extremes of the 3-5 m bandpass filter of L* (Furlo), MS (Bottaccione), as well as and on the stacking pattern of shales and cherts (Bottaccione).

For the Livello Bonarelli from Furlo (124 cm), a duration estimate is obtained from 2-cm spaced X-Ray Fluorescence (XRF) spectrometry data. The multitaper method (MTM) spectral analyses of $SiO_2$ yield dominant periodicities of ~40, ~12, and ~6 cm (Fig. 3a). We calculate the ASM using the results of MTM harmonic analysis (>80%), and obtain an optimal sedimentation rate of 0.286 cm/kyr for the Bonarelli in Furlo. Hence, we interpret the reported periodicities as the imprint of short eccentricity, obliquity, and precession and estimate the duration of the Livello Bonarelli at 413 kyr.

ASM analysis of the $Al_2O_3$ data from the 82 cm thick Livello Bonarelli at Bottaccione suggests an optimal sedimentation rate of 0.208 cm/kyr (Fig. 5). The ~8-cm thick cycles are interpreted as the imprint of obliquity and the duration of Livello Bonarelli at Bottaccione is estimated at 410 kyr, comparable to the estimate of 413 kyr at Furlo.

## 4    Discussion

### 4.1    Proxy records and correlation of the C/T boundary

The records of $\delta^{18}O$ and $\delta^{13}C$ show long-term trends over the successions, although the $\delta^{18}O$ record in particular displays scatter, which might be due to an influence of diagenesis. The $\delta^{13}C$ signal is generally more robust to post-depositional alteration (Jenkyns et al., 1994) and bulk carbonate $\delta^{13}C$ patterns constitute a powerful tool for stratigraphic correlation, despite variations in absolute values and amplitude amongst locations (Jarvis et al., 2006). The Cenomanian record presented here displays a higher degree of variability than a recently published bulk carbonate $\delta^{13}C$ record from Furlo by Gambacorta et al. (2015). The high variability in $\delta^{13}C$ values from 476 to 484 m coincides with a frequent occurrence of organic-matter rich beds, which may have influenced $\delta^{13}C$ values of early diagenetic cements. Although variability at the sampling scale (3 cm) may partially represent effects of diagenesis which could obscure short (precessional scale) climatic signals, the longer term trends

compare well with coeval sections in the Umbria-Marche basin (Sprovieri et al., 2013; Stoll and Schrag, 2000) and the English chalk records (Jarvis et al., 2006) (Fig. 9; see Section 4.5).

Whereas the stable isotope data reflect variations in temperature, salinity and the global carbon cycle, with superimposed regional and diagenetic effects, the other proxy data are closely related to lithological variations. These lithological variations reflect differing contributions of detrital input (indicated by magnetic susceptibility, $Al_2O_3$, $TiO_2$, $SiO_2$) and biological productivity of organic matter, carbonate and biogenic silica (reflected in the colour reflectance, LOI and $SiO_2$ records). These variations show a local and direct response to orbitally-forced variations in temperature, run-off and ventilation.

In this study, the base of the Turonian stage at Bottaccione is placed at 487.47 m, just above the first occurrence of *Quadrum gartneri* at 487.25 m defining the base of the C11 zone (Sissingh, 1977), following Sprovieri et al. (2013) and Tsikos et al. (2004). Direct comparison of high resolution $\delta^{13}C_{carb}$ data from Pueblo (Caron et al., 2006) with $\delta^{13}C_{carb}$ data from Contessa (Stoll and Schrag, 2000) allows for correlating $\delta^{13}C_{carb}$ maximum III in Pueblo with the $\delta^{13}C_{carb}$ maximum 85 cm above the Livello Bonarelli at 487.52 cm, comparable with its location at Bottaccione. An alternative correlation places the C/T boundary near a minimum in the $\delta^{13}C_{carb}$ data from the Gubbio S2 core (Trabucho-Alexandre et al., 2011) of Tsikos et al. (2004), which corresponds to the minimum in $\delta^{13}C_{carb}$ values at 488.22 m at Contessa (Stoll and Schrag, 2000), which is 75 cm above the C/T estimate adopted in this study. Consequently, 75 cm is taken as a conservative estimate of the stratigraphic uncertainty on the position of the C/T boundary, i.e. 487.47 ± 0.75 m. With an average sedimentation rate of 11 m/Myr, this stratigraphic error margin translates in a temporal uncertainty of ± 68 kyr.

## 4.2 Astronomical forcing and calibration

### 4.2.1 Astronomical phase relations

Throughout the Cenomanian interval of the Furlo section, black cherts occur in distinct bands, which are often underlain by a thin layer of black shale. As these organic-rich horizons do not occur as nodules, they reflect a primary silica enrichment from radiolarian and/or diatom blooms. When present, black chert bands occur in groups with a regular spacing amongst them, likely reflecting a threshold response to extremes of the precessional cycle. Previous tuning attempts have placed black cherts either in eccentricity maxima (Mitchell et al., 2008; Voigt et al., 2006) or eccentricity minima (Lanci et al., 2010), entailing distinctly different

oceanographic regimes. During eccentricity maxima, the seasonal contrast on the Northern Hemisphere is periodically enhanced during high-amplitude precession minima, thereby intensifying monsoons, leading to an estuarine circulation in the Cretaceous North Atlantic with upwelling and increased productivity (Mitchell et al., 2008), potentially spurred by input of nutrients from volcanic activity (Trabucho-Alexandre et al., 2010). Alternatively, it has been suggested that eccentricity minima could cause decreased seasonality, leading to stagnation and reduced ventilation of bottom waters (Lanci et al., 2010; Herbert and Fischer, 1986), although eccentricity minima would not lower seasonality but rather avoid large seasonal extremes for a prolonged period of time. This reverse phase relationship is deduced from the remanent magnetization within carbonates at Furlo (Lanci et al., 2010), unfortunately excluding cherts and thereby obscuring the imprint of precession cycles on the sedimentary rhythms. By analysis of frequency modulation on the same dataset, Laurin et al. (2016) re-evaluated the phase relationship and concluded that periods of increased black chert deposition coincided with eccentricity maxima.

The relationship between eccentricity forcing and ocean-climate response can be derived from the degree of variability in the presented data. Intervals marked by maximal lithological difference represent periods of large precessional amplitude during eccentricity maxima. Radiolarian cherts coincide with maximal amplitude of carbon and oxygen isotope signals and with a tendency of those proxies towards more negative values (Figs. 2 & 3). Negative $\delta^{18}O$ values may reflect warmer temperatures and a potentially increased influx of fresh water by increased monsoonal activity. Relatively low values of $\delta^{13}C$ could be associated with stratification of the water column and reduced yearly-integrated primary productivity (Sprovieri et al., 2013). Conversely, high $\delta^{13}C$ values likely reflect good bottom water ventilation during eccentricity minima, with a prolonged avoidance of seasonal extremes, allowing for more stable primary productivity over the annual cycle that may have caused the increase in marine $\delta^{13}C$ (Fig. 3). Possibly, the increased accumulation of organic carbon on land due to more uniform annual precipitation during eccentricity minima may have amplified the rise in marine $\delta^{13}C$, as suggested for Cenozoic intervals (Zachos et al., 2010). Figure 2 illustrates this phase relationship based on proxy records from the Cenomanian Furlo section. An analogous phase relationship for the proxy records from the Bottaccione sections is inferred. There, black cherts are absent from the Turonian interval of the succession, but grey cherts occur rhythmically throughout. Increased variability and negative values of $\delta^{13}C$

coincide with high variability in the magnetic susceptibility record in chert-rich intervals,
associated with eccentricity maxima.

### 4.2.2   Calibration to 405-kyr eccentricity

In this study, we distinguish minima and maxima of the 405-kyr eccentricity cycle within the
Scaglia Bianca and Scaglia Rossa by examining the band-pass filters of the geophysical
proxies. Near the end of the dataset in Furlo, below the Livello Bonarelli, and for the
Turonian interval above the Livello Bonarelli, the pattern of individual limestone-chert
alternations is taken into account instead. The band-pass filters of the stable isotope data are
presented to evaluate the cyclostratigraphic framework.
As mentioned above, multiple previous tuning efforts have been published on the
Cenomanian-Turonian sections in the Umbria-Marche. However, the present study provides a
clear advancement over previous reports because: (i) we use only the 405-kyr periodicity of
eccentricity in the La2011 solution as tuning target; (ii) we present an independent estimate
for the timespan from the base of the Livello Bonarelli to the C/T boundary; (iii) we use the
calibrated age of 28.201 Ma (Kuiper et al., 2008) for the Fish canyon sanidine standard for
$^{40}$Ar/$^{39}$Ar dating; and (iv) we provide a new radioisotopic age for the mid-Cenomanian event.
We discuss each of these aspects in the following paragraphs.
We correlate interpreted 405-kyr eccentricity minima in the lithology and geophysical data to
405-kyr minima in the La2011 (nominal) eccentricity solution (Laskar et al., 2011a), obtained
by band-pass filtering (300-625 kyr). Only the 405-kyr component of eccentric is stable
beyond 50 Ma, and it is the prime tuning target for the Cretaceous. The shorter obliquity and
eccentricity-modulated precession terms can only be used for the development of floating
time scales. Previous tuning efforts have used the ~100 kyr periodicity of eccentricity
(Mitchell et al., 2008), extracted from the La2004 solution, which is only considered reliable
until 40 Ma (Laskar et al., 2004).
Two 405-kyr tuning options to astronomical solution La2011 remain if a C/T boundary age of
93.9 ± 0.15 Ma (Sageman et al., 2006; Meyers et al., 2012b) is considered, along with
stratigraphic uncertainty in the studied sections (± 0.068 Ma) and uncertainty in the
astronomical solution. The uncertainty in the 405-kyr component of the astronomical target
curve was estimated at ± 78.5 kyr, by determining the maximal difference between the
position of minima in the 405-kyr band-pass filter outputs (300-625 kyr) of the La2010d,
La2011 and La2011m2 solutions (Laskar et al., 2011a). The first 405-kyr minimum in the
Turonian at 489.1 m in Bottaccione corresponds to the 405-kyr minimum in the astronomical
solution at 93.5 ± 0.15 (tuning #1) or at 93.9 ± 0.15 Ma (tuning #2; Fig. 3).
Tuning options for the Cenomanian interval of this study depend on the duration of the
Livello Bonarelli. The Livello Bonarelli in the Umbria-Marche basin reflects the culmination
of OAE2, limited to the "2$^{nd}$-build-up" and "plateau" of the OAE2 $\delta^{13}$C excursion (Tsikos et
al., 2004). The estimated duration in this study between the start of the $\delta^{13}$C excursion, below
the Livello Bonarelli, and the Cenomanian-Turonian boundary is ~490 kyr. This duration is
slightly longer than a previous estimate from the German Wunstorf core of 430-445 kyr for
the OAE2 isotope excursion (Takashima et al., 2009; Voigt et al., 2008), and slightly shorter
than the duration of of 520 – 560 kyr from the "first build-up" to the "end of plateau",
determined by intercalibration between radioisotopic and astrochronologic time-scales at the
C/T GSSP (Sageman et al., 2006; Meyers et al., 2012b). Similar duration estimates for this
interval were obtained by reinterpreting the orbital influence at Demerara Rise and Tarfaya
(Meyers et al., 2012a), of 500-550 kyr and 450-500 kyr respectively, as well as in the
Aristocrat-Angus-12-8 Core in Northern Colorado (Ma et al., 2014), of 516-613 kyr, and the
Iona Core in Texas of ~540 kyr (Eldrett et al., 2015), albeit using slightly different
correlations.
A potential complication arises from the sharp shifts in sedimentary facies at the base and the
top of the Livello Bonarelli, which could be accompanied by hiatuses. A hiatus on the order
of 20 kyr at the base of the black shale has been suggested by Jenkyns et al (2007). Such a
hiatus would be relatively small compared to our tuning target, the 405-kyr periodicity of
eccentricity-modulated precession. In contrast, Gambacorta et al. (2015) suggest a large hiatus
near the top of the Livello Bonarelli, based on correlation of the phases of OAE2. This view is
considered unlikely in this study, as there is no strong sedimentary expression of such a
hiatus, the duration of black shale deposition estimated here is in good agreement with other
studies and the first occurrence of *Quadrum gartneri* is detected 58 cm above the base of the
Scaglia Rossa.
The duration estimate for the Livello Bonarelli allows the extension of the astronomical
tuning into the Cenomanian interval of this study. The base of the Livello Bonarelli
corresponds to 94.19 ± 0.15 Ma (tuning #1) or 94.59 ± 0.15 Ma (tuning #2); i.e., the first
short-eccentricity maximum after a 405-kyr minimum.

### 4.2.3  Integration with radioisotopic ages

The age for the base of the Livello Bonarelli of 94.19 ± 0.15 Ma (tuning #1) is similar to the previously reported age of 94.21 Ma (Mitchell et al., 2008). Nonetheless, Mitchell et al. (2008) used radioisotopic ages of Sageman et al. (2006), which were calculated using an age of 28.02 ± 0.28 Ma for the Fish Canyon sanidine standard (Renne et al., 1998), widely used in $^{40}Ar/^{39}Ar$ dating. In this study, ages are calculated with the recalibrated age of the Fish Canyon sanidine of 28.201 ± 0.046 Ma (σ) by Kuiper et al. (2008). Recalibration of the reported tuned age of 94.21 Ma (Mitchell et al., 2008) would correspond to an age of 94.81 Ma after recalibration to the revised standard age of Kuiper et al. (2008), a difference of 1.5 x 405-kyr cycle.

Additional age control is provided by correlation of two Cenomanian ash beds from the Western Interior of the USA. Correlation to "Ash A" at the base of the boundary of the planktonic foraminifer biozones of *Whiteinella archaeocretacea* and *Rotalipora cushmani* (Sageman et al., 2006; Caron et al., 2006; Leckie, 1985) provides an independent age for this zonal boundary 7 cm below the base of the Bonarelli Level of 94.20 ± 0.28 Ma. This age is in closer agreement to tuning #1 (94.17 ± 0.15 Ma) than to tuning #2 (94.57 ± 0.15 Ma).

The MCE, characterized by a double positive peak in $\delta^{13}C$ at Furlo (first maximum at 466.47 m), offers another opportunity to test both tuning options. The $^{40}Ar/^{39}Ar$ isotope data were acquired from 41 single sanidine crystals in sample 91-O-03 of Obradovich (1993) (methods outlined in Sageman et al., 2014), and yields an age of 96.21 ± 0.16/0.36 (2σ analytical and full uncertainty) for the Thatcher bentonite in the *Conlinceras tarrantense* zone at Pueblo, Colorado. This bentonite falls within the first peak of the MCE (Gale et al., 2008). In our tuning options, this level is either 96.09 ± 0.15 Ma (tuning #1) or 96.49 ± 0.15 Ma (tuning #2). Although tuning #2 is in better agreement with Eldrett et al. (2015), who reports an age for the onset of the OAE2 carbon isotope excursion of 94.64 ± 0.12 Ma, the correlation to radioisotopic ages leads us to favour the first tuning option. Tuning #1 is in close agreement with the new $^{40}Ar/^{39}Ar$ age for the mid-Cenomanian event, the intercalibrated age for Ash A at the base of the *Whiteinella archaeocretacea* zone and the age of the C/T boundary as determined by Meyers et al. (2012b). Nonetheless, the duration between radioisotopic age tie-points is consistent with cyclostratigraphy (Fig. 2) and provides tuned ages for biostratigraphic events (Table 1).

## 4.3 Long term behaviour of the carbon cycle

### 4.3.1 Expression of long-term eccentricity forcing

Superimposed on the hierarchical stacking patterns of lithologies in the studied succession, several features of the lithological and proxy records reveal the influence of long-term periodicities on local sedimentation and global climate. These observations include: (i) the absence of cherts in an interval below the Livello Bonarelli; (ii) a strong expression of obliquity forcing during deposition of the Livello Bonarelli, contemporaneous with a sedimentary response to the 100-kyr forcing of eccentricity; (iii) a spacing of 2.0 and 2.4 Myr, respectively, between the mid-Cenomanian $\delta^{13}C$ excursion, the onset of OAE2, and a positive $\delta^{13}C$ excursion in the mid-Turonian. These observations, in combination with a previously noted ~1 Myr cyclicity in $\delta^{13}C$, reveal a pacing of climatic events by long-term eccentricity cycles and will be further discussed in the following paragraphs.

Below the Livello Bonarelli, in the interval 483-485 m, black shales are conspicuously absent. This may partially be due to an increase in sedimentation rate, as indicated by a larger spacing between beds from 483.5 m upwards, but this pattern breaks the trend of an increasing number of black cherts and shales up-section, per meter as well as per interpreted ~100 kyr bundle. This may reflect the prolonged avoidance of seasonal extremes during long-term eccentricity minima of the 2.4-Myr cycle. The first ~100-kyr bundle of black cherts following this interval contains exceptionally thick dark levels and corresponds to the beginning of the first 405-kyr maximum after the ~2.4-Myr minimum. Hence, we associate the onset of OAE2 with this 405-kyr maximum.

Within the Livello Bonarelli, the imprint of 100-kyr eccentricity cycles can be observed, comparable to the expression of OAE2 in the Sicilian Calabianca section (Scopelliti et al., 2006) and in the German Wunstorf core (Voigt et al., 2008). Additionally, ten obliquity-related cycles can be visually detected in the XRF-proxy data (Fig. 4b-e). Silica, delivered by radiolarian blooms, mirrors terrestrially derived components ($Al_2O_3$ and $TiO_2$) and may represent variations in seasonality and ventilation driven by obliquity during the deposition of the Livello Bonarelli.

Two pronounced 405 kyr minima, likely within a 2.4 Myr minimum, occur in the upper Cenomanian, the first of which could correspond (following tuning #1) to the interval lacking black shales at 483-485 m, and the second occurring within the Livello Bonarelli. The

occurrence of two 405 kyr minima within a 2.4 Myr minimum could explain the observed presence of the ~100 kyr cyclicity within the Livello Bonarelli, as well as the influence of obliquity, also detected during OAE2 in several North Atlantic datasets (Meyers et al., 2012a). The Livello Bonarelli was previously suggested to coincide with a 2.4 Myr eccentricity minimum, invoking stagnation as forcing mechanism for anoxia (Mitchell et al., 2008). The relatively strong obliquity influence during the deposition of the Livello Bonarelli is consistent with this orbital configuration (Hilgen et al., 2003).

The new astrochronologies allow for assessing the long-term behaviour of the carbon cycle during the C/T transition. The onset of the MCE, the base of the Livello Bonarelli, and the middle of the negative $\delta^{13}C$ excursion of the mid-Turonian are separated by 2.0 Myr and 2.4 Myr, respectively. The 1.6 Myr long negative excursion in the mid-Turonian is characterized by an intermittent double positive peak ("Pewsey events"; Jarvis et al., 2006; Fig. 9), similar to the MCE, starting at 91.7 Ma (tuning #1) or 92.1 Ma (tuning #2). These repetitive variations in $\delta^{13}C$ are likely paced by the ~2.4-Myr eccentricity period. Following tuning #1, a tentative comparison with the full eccentricity solution La2011 (Fig. 2) reveals the occurrence of pronounced long-term minima in eccentricity before the mid-Cenomanian and mid-Turonian events. An influence of long-term (several Myr) cycles on $\delta^{13}C$ has been previously identified in a late Cretaceous $\delta^{13}C$ record from Bottaccione (Sprovieri et al., 2013). Recently, a ~1 Myr cycle was detected in the long term behaviour of $\delta^{13}C$, particularly in the Turonian record from the Bohemian Cretaceous Basin (Fig. 9), and attributed to the ~1.2 Myr cycle in amplitude modulation of Earth's axial obliquity (Laurin et al., 2015). Although a ~1 Myr periodicity cannot be identified in our data, an influence of obliquity forcing is observed in the Cenomanian part of the record. Sharp positive excursions on a ~2.4 Myr scale may have occurred superimposed on gradual ~1 Myr cycles in $\delta^{13}C$ variation.

The Cenomanian $\delta^{13}C$ curve is more strongly paced by the 405 kyr cycle than the Turonian $\delta^{13}C$ curve. Such a change was previously observed for the end of the Albian and interpreted to reflect a change to more stable ocean circulation patterns (Giorgioni et al., 2012). For the Cenomanian/Turonian, the carbon cycle may have become more stable as CO2 was drawn down by organic matter deposition and volcanic activity decreased.

**4.3.2   Relation with volcanism**

The increasing recurrence of black cherts through the Cenomanian interval indicates that the western Tethys was progressively more prone to the development of anoxia due to a long-

term trend of on-going warming and increased volcanism. Trace element studies point to volcanic activity at the Caribbean Large Igneous Province as the supplier of nutrients and sulphate to a low-sulphate ocean, with a major pulse ~500 kyr before OAE2 (Snow et al., 2005; Turgeon and Creaser, 2008; Adams et al., 2010; Jenkyns et al., 2007). Volcanism is thus ultimately responsible for OAE2, but the exact timing of the onset of OAE2 seems to be linked to a specific sequence of astronomical variations superimposed on this trend. The increased variability in seasonality, after the prolonged avoidance of seasonal extremes, gave rise to an intensification of the hydrological cycle, weathering, and more vigorous ocean circulation, which is in agreement with several Nd-isotope records (Martin et al., 2012; Zheng et al., 2013) and Os isotope records (Du Vivier et al., 2014). Deep waters at Demerara Rise were replaced by bottom waters sourced from the Tethys and North Atlantic (Martin et al., 2012). Trabucho-Alexandre et al. (2010) suggest that the OAE2 interval may have been characterised by an intense estuarine circulation with upwelling in the proto-North Atlantic. This is consistent with the phase relationship inferred from our data: with black chert and shale deposition coincident with seasonality extremes during 405-kyr eccentricity maxima. Previously, Mitchell et al. (2008) placed OAE2 within the ~2.4 Myr minimum itself and suggested that the lack of strong insolation variability, associated with such a minimum, prevented the system from changing states and hindered limestone deposition. However, our chronology advocates intensified circulation and upwelling, delivering nutrients from volcanism and weathering to the western Tethys and the North Atlantic, and triggering prolonged and widespread anoxia. In conclusion, the 6-Myr-long astronomically-tuned timescale across OAE2 presented in this study allows for the evaluation and combination of two leading hypotheses about OAE2 forcing mechanisms.

**Acknowledgements**

Special thanks go to the Association "Le montagne di San Francesco" for logistic support in the field, as well as the inhabitants of Coldigioco. We would like to thank Jiří Laurin and two anonymous reviewers for their feedback, as well as Stephen Meyers, Nicolas Thibault, Ian Jarvis, André Bornemann and an anonymous reviewer for comments on a previous version of the manuscript. DDV thanks the Research Foundation Flanders (FWO) for a Ph.D. scholarship and the EARTHSEQUENCING project (ERC Consolidator Grant). SJB thanks the European Community funded GTSnext project (grant agreement 215458), the European

Science Foundation activity 'EARTHTIME—The European Contribution' (Exchange Grant
3818) and the German Research Foundation (DFG VO 687/14-1, IODP/ODP SSP 527/32)
PhC acknowledges the support of the Hercules foundation for upgrade of the VUB stable
isotope laboratory.

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

**Table 1.** Astronomical tuning options for biostratigraphic and isotopic events and comparison to radioisotopic ages. Uncertainties on tuned absolute ages comprise the uncertainty in the stratigraphic position and/or correlation of an event (± 0.75 m) and the uncertainty in the astronomical target curve (Laskar et al., 2011a) (± 0.079 Myr). The numerical age of the C/T boundary is based on intercalibration of $^{40}Ar/^{39}Ar$ dating, U-Pb dating and astrochronology (Meyers et al., 2012b); Radioisotopic age of the Base of *Whiteinella archaeocretacea* is the weighted mean age of single and multicrystal $^{40}Ar/^{39}Ar$ ages of bentonite A; Radioisotopic ages for the first peak of the Mid-Cenomanian event come from single crystal $^{40}Ar/^{39}Ar$ dating of the Thatcher bentonite in the *Conlinceras tarrantense* zone (Calycoceras *gilberti*). All radioisotopic ages are reported in 2σ and using a FC age of 28.201 ± 0.046 ka (1σ) (Kuiper et al., 2008).

| Event | Stratigraphic Level (m) | Tuning #1 (Ma) with stratigraphic and astronomical uncertainty | Tuning #2 (Ma) with stratigraphic and astronomical uncertainty | Radioisotopic dating (Ma) with 2σ radiometric uncertainty |
|---|---|---|---|---|
| *Hitch Wood* δ$^{13}$C excursion | 516.24 m | 90.59 ± 0.15 | 90.99 ± 0.15 | |
| **Base D. primitiva - M. sigali** | 508.85 m | 91.31 ± 0.15 | 91.72 ± 0.15 | |
| *Round down* δ$^{13}$C excursion | 499.44 m | 92.32 ± 0.15 | 92.72 ± 0.15 | |
| **Base NC14** | 493.85 m | 92.93 ± 0.15 | 93.33 ± 0.15 | |
| **Base *Helvetoglobotruncana helvetica*** | 491.85 m | 93.17 ± 0.15 | 93.57 ± 0.15 | |
| **C/T boundary** | 487.47 m | 93.69 ± 0.15 | 94.10 ± 0.15 | 93.90 ± 0.15 [Meyers et al., 2012b] |
| **Base *Whiteinella archaeocretacea*** | 485.70 m | 94.17 ± 0.15 | 94.57 ± 0.15 | 94.20 ± 0.28 [Meyers et al., 2012b] |
| **Base NC12** | 484.20 m | 94.28 ± 0.15 | 94.68 ± 0.15 | |
| **Base CC10** | 482.77 m | 94.39 ± 0.15 | 94.79 ± 0.15 | |
| **first peak Mid-Cenomanian event** | 466.47 m | 96.09 ± 0.15 | 96.49 ± 0.15 | 96.21 ± 0.36 [this study] |

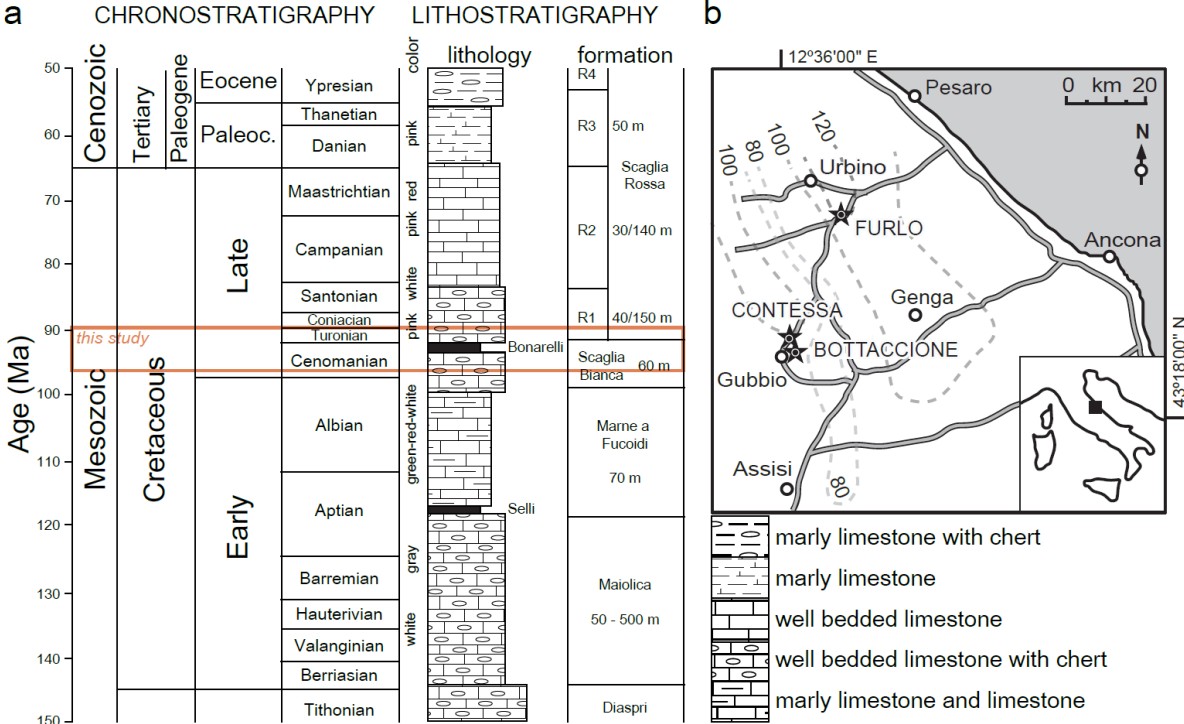

Figure 1. **(a)** Chrono- and lithostratigraphy of the Umbria-Marche succession in the Apennine mountains (adapted from Alvarez et al., 2011). **(b)** Geographic setting of the study sections with isopachs of the Bonarelli level (in cm) from Montanari et al.(1989).

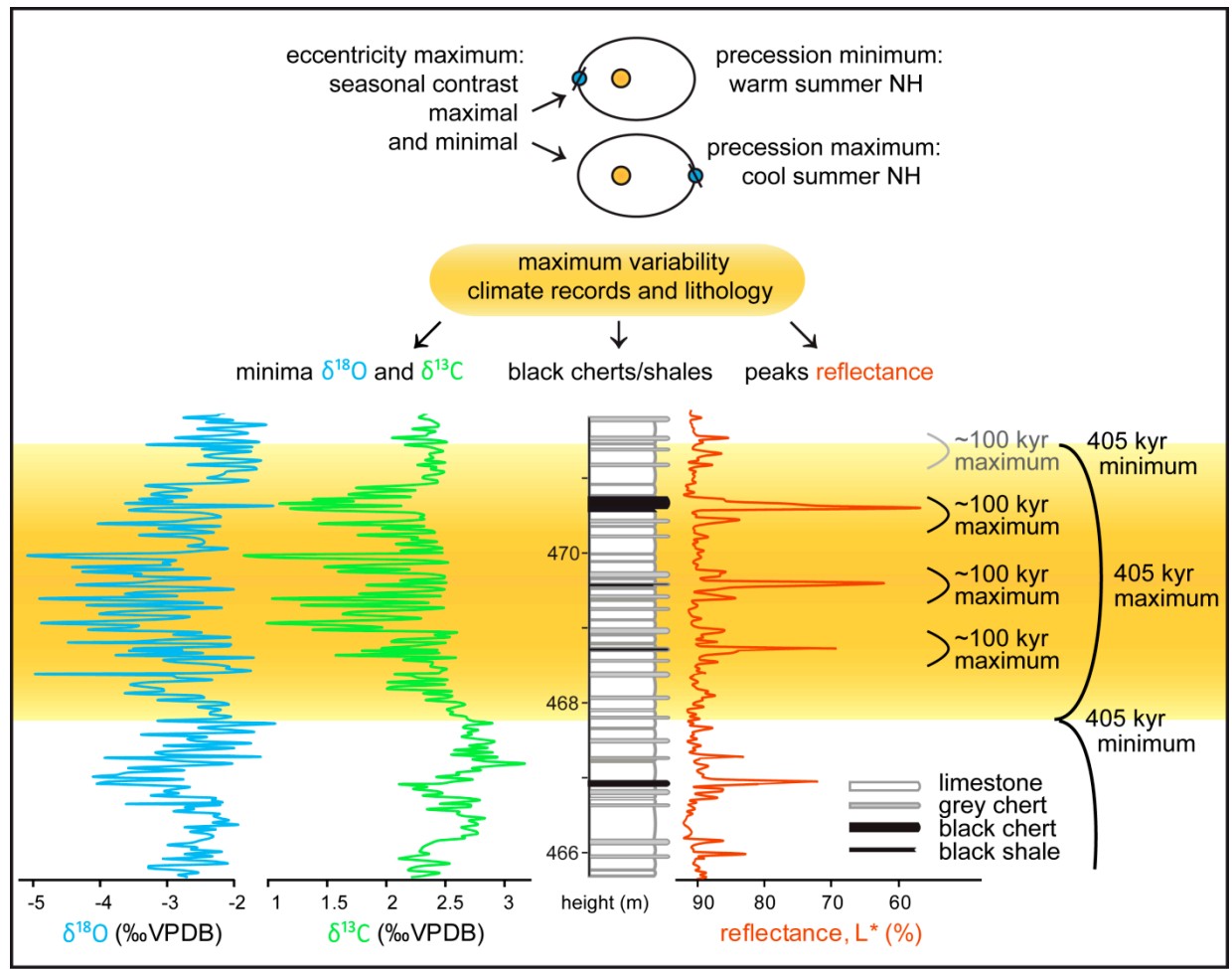

Figure 2. Phase relationship between eccentricity and proxy records. During eccentricity
maxima, the seasonal contrast for the NH is maximally enhanced during precession minima
and maximally reduced during precession maxima (top: schematic representation of the
Earth's orbit around the sun). Hence, climate variability is strongly amplified during
eccentricity maxima, triggering the highest variability in the climate-sensitive records and
hierarchical organization of chert-limestone alternations.

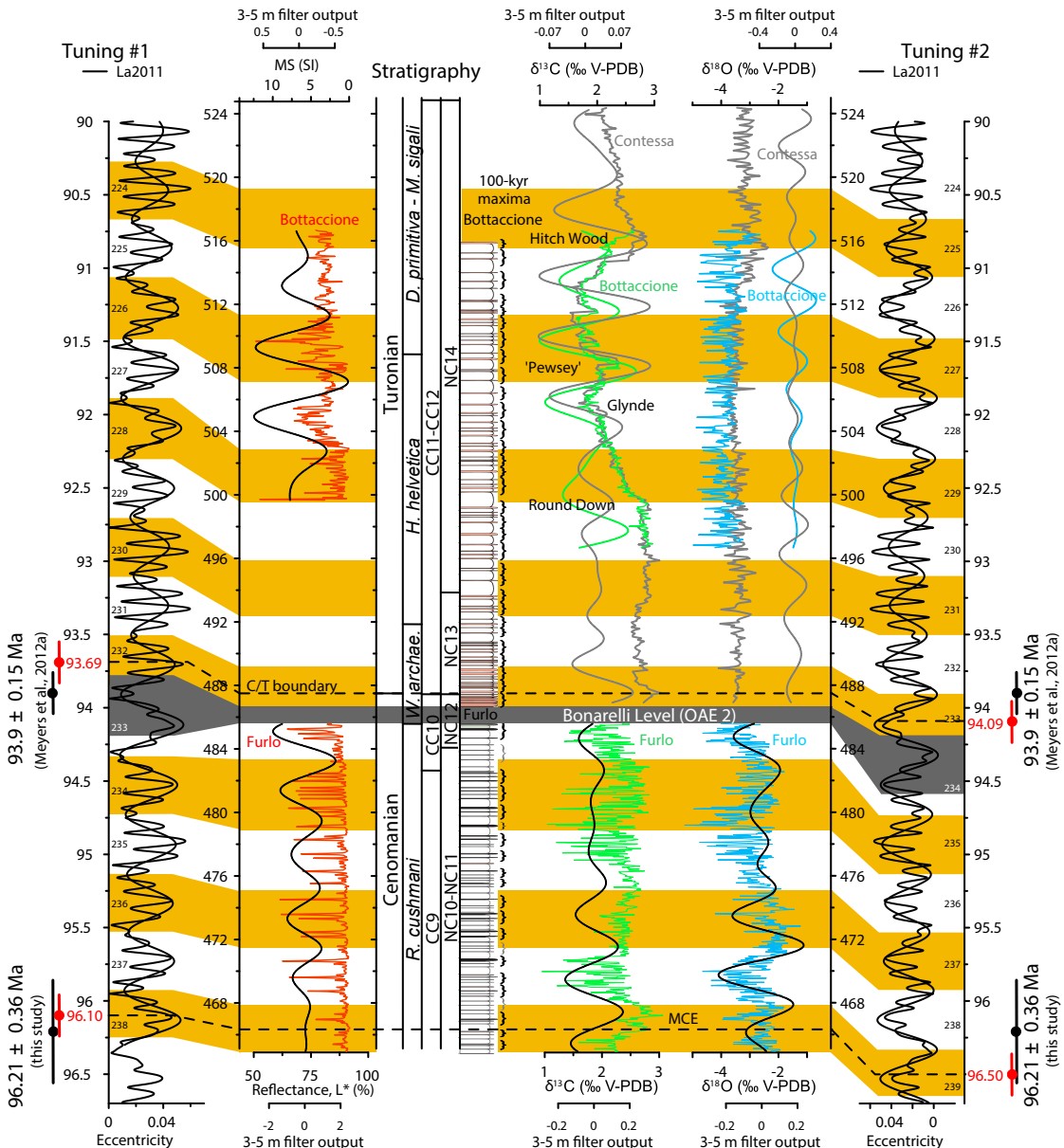

Figure 3. Cyclostratigraphic interpretation of the C/T interval of the Umbria-Marche basin. Brackets indicate 100-kyr bundles of precession-paced lithological alternations, further grouping in 405-kyr cycles is indicated by alternating yellow-white bands. Geophysical records (L* and MS) accentuate the hierarchical stacking pattern. Stable isotope ratios show increased amplitude and a tendency towards more negative values during eccentricity maxima. Isotopic records in grey are from Stoll and Schrag (2000). Event names in italics refer to nomenclature of Jarvis et al. (2006). Astronomical tuning options to La2011 are presented, with 405-kyr cycle numbering back from the present day, next to radioisotopic ages discussed in this study.

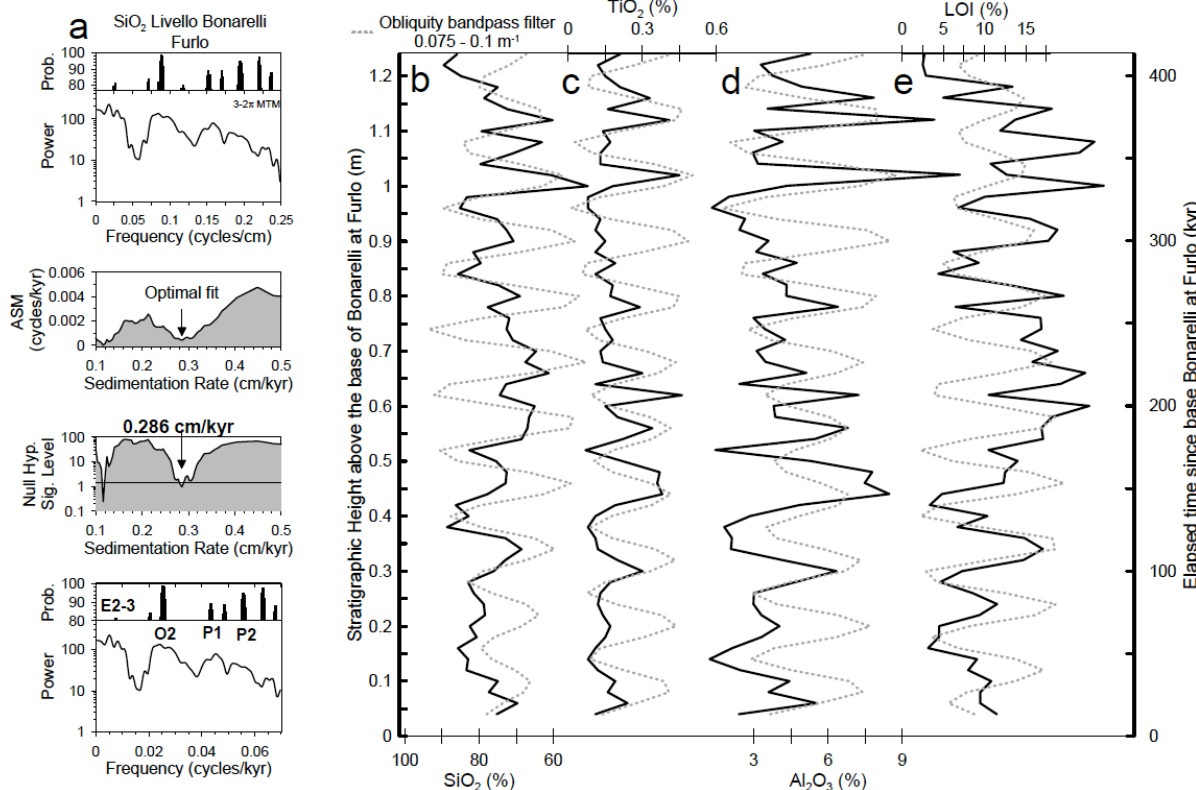

Figure 4. Duration estimate of Livello Bonarelli at Furlo based on **(a)** the Average Spectral Misfit (ASM) method. **(b-e)** SiO₂, TiO₂, and Al₂O₃ contents and Loss on Ignition (LOI) data show ~12-cm-thick cycles, interpreted as obliquity. The duration of Livello Bonarelli at Furlo is estimated at 413 kyr.

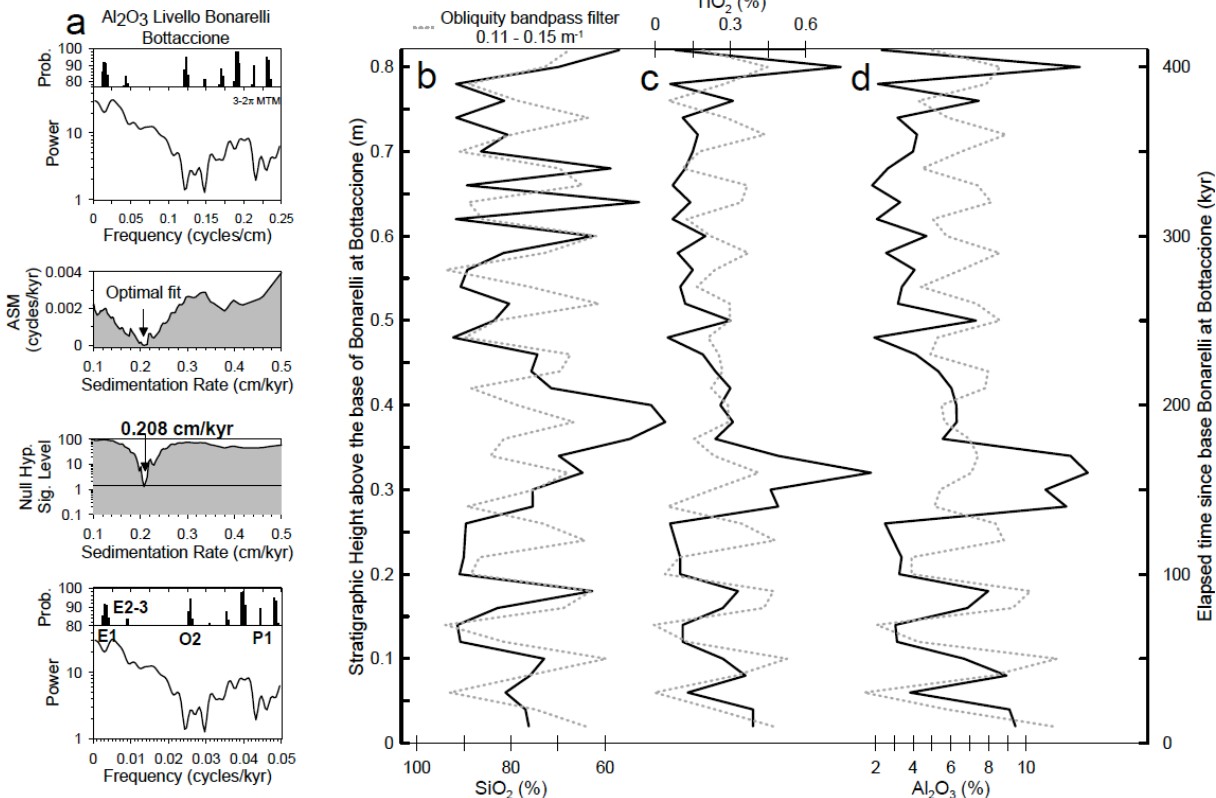

Figure 5. Duration estimate of Livello Bonarelli at Bottaccione based on (a) the Average
Spectral Misfit (ASM) method. (b-d) $SiO_2$, $TiO_2$ and $Al_2O_3$ data show ~8-cm thick cycles,
interpreted as obliquity. The duration of Livello Bonarelli at Bottaccione is estimated at 410
kyr.

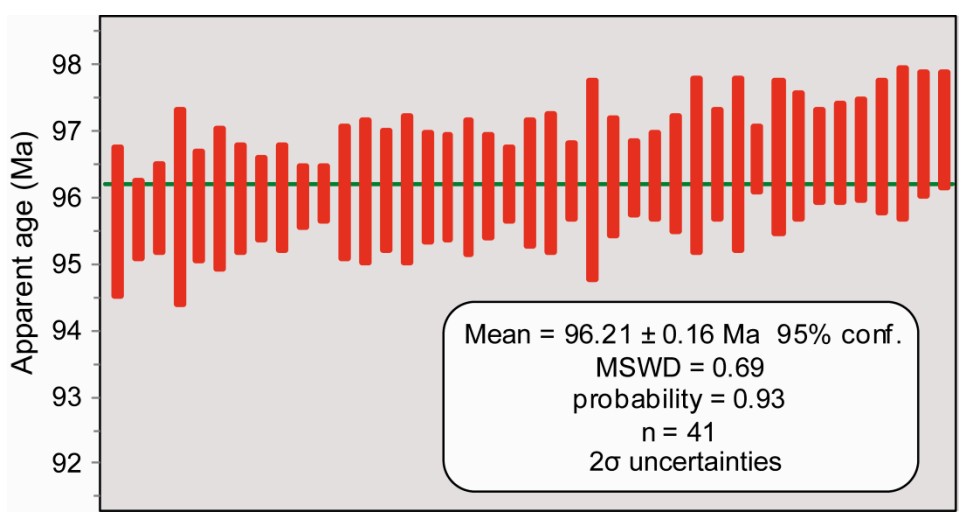

2 Figure 6. Summary of $^{40}$Ar/$^{39}$Ar results

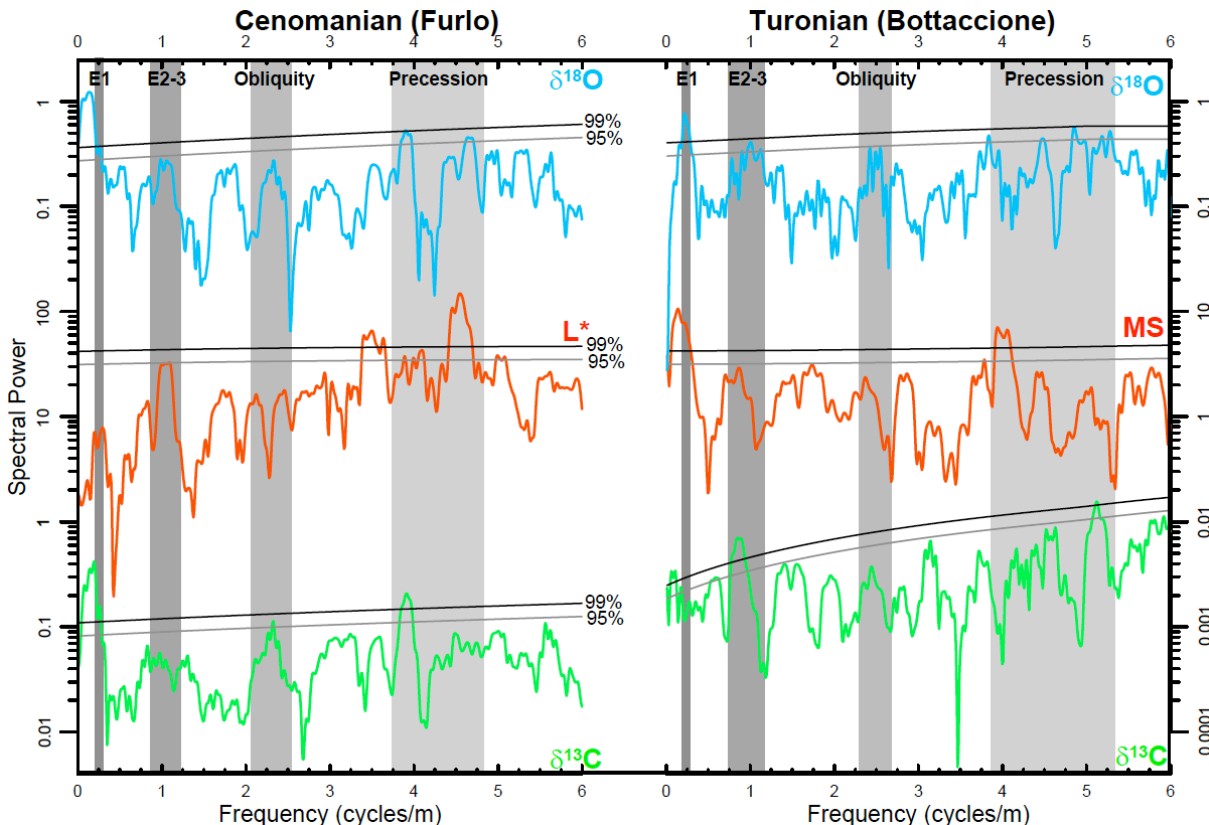

2    Figure 7. MTM/LOWSPEC spectra of proxy records. All proxy records show a strong imprint

3    of eccentricity-modulated precession (E2-3: short eccentricity), $\delta^{13}C$ from Furlo also displays

4    a statistically significant (>95% confidence level) imprint of obliquity.

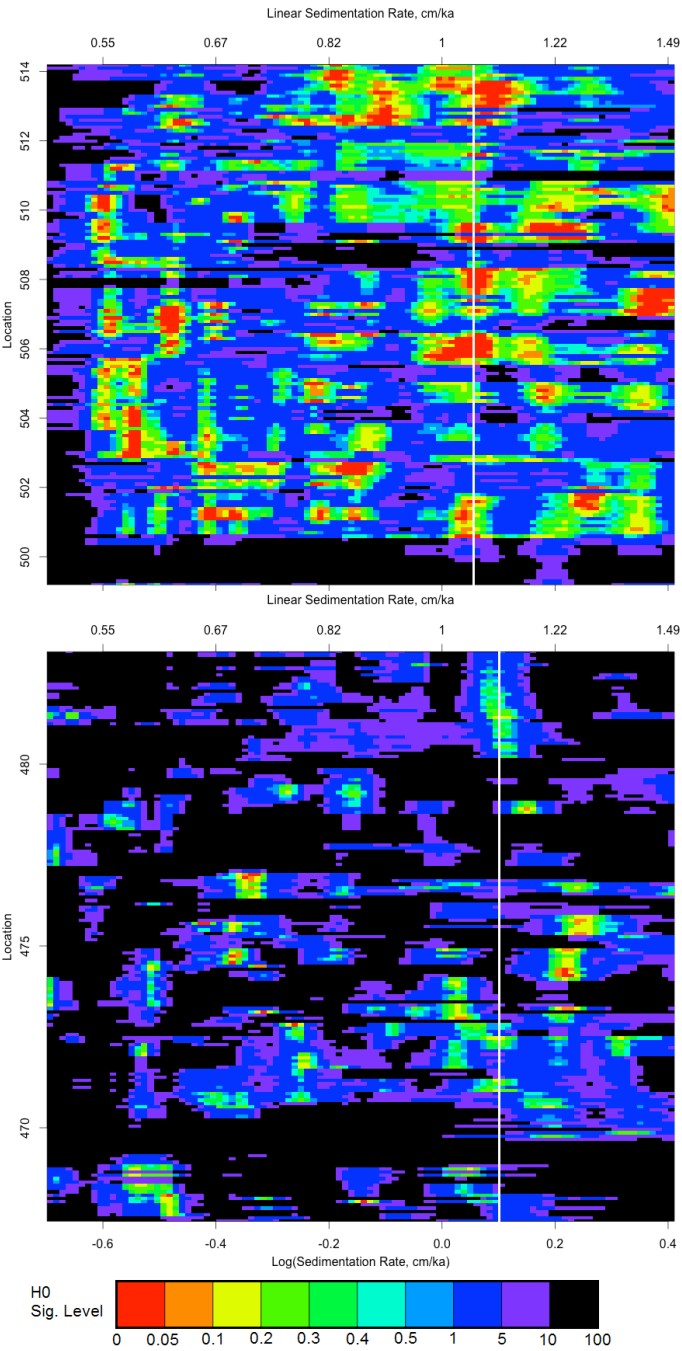

Figure 8. Evolutionary average spectral misfit (eASM) of the $\delta^{13}C$ data from Furlo (bottom)
and Bottaccione (top), with a 5 m window, 0.1 m steps and using those frequencies with F-
test > 80%. The white line suggests a stable sedimentation rate of 1.1 cm/kyr in Furlo and
1.05 cm/kyr in Bottaccione.

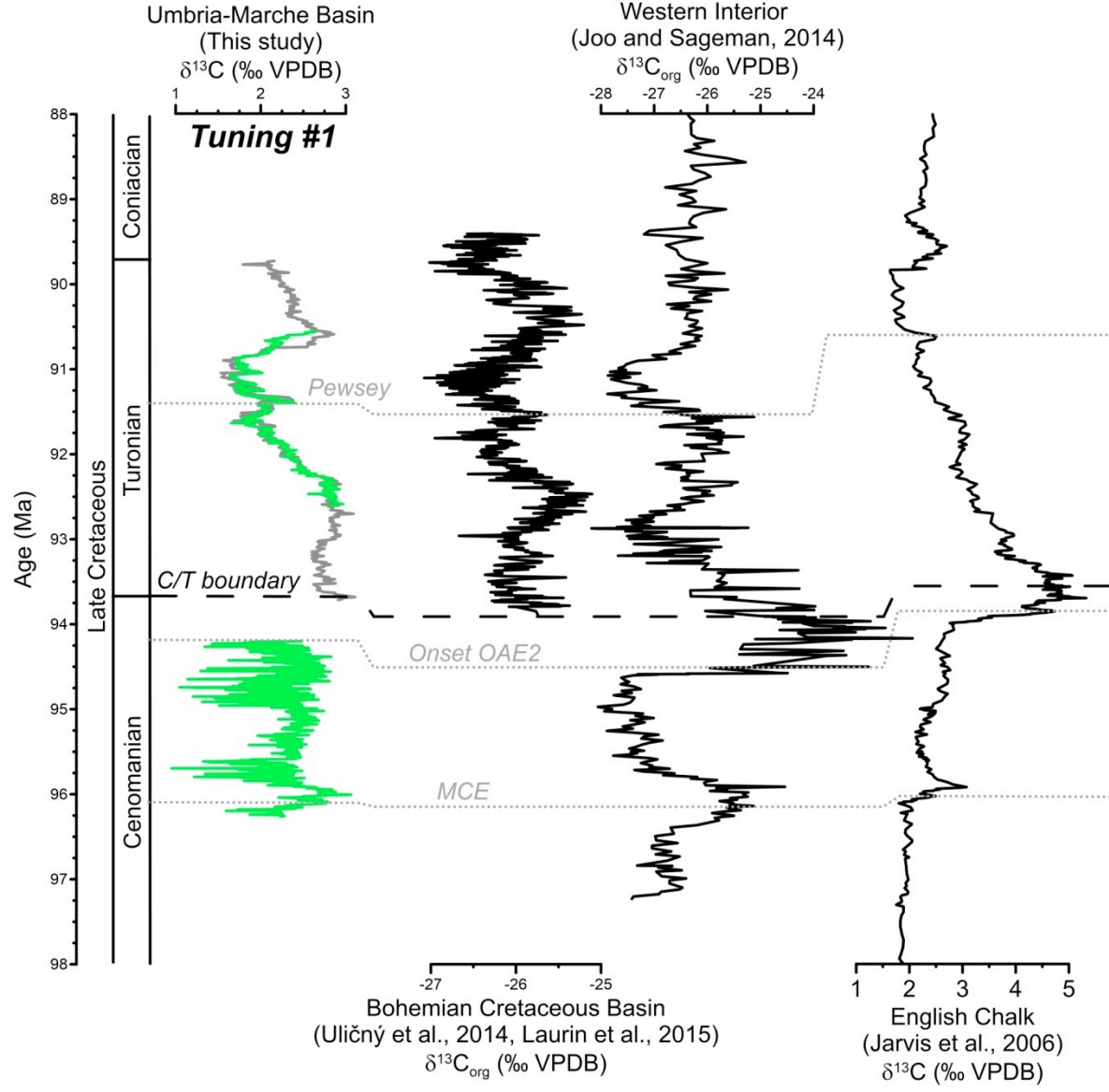

Figure 9. Global correlation of $\delta^{13}C_{carb}$ data of the Cenomanian-Turonian interval.