# Peer review of "Orbital control on the timing of oceanic anoxia in the Late"

_Climate of the Past, 2015_

## Referee Comment (RC1) · Anonymous Referee #1 · 28 Feb 2016

1. Does the paper address relevant scientific questions within the scope of CP? Yes, the authors want to show and explain climate control on the development of poor oxygenation conditions in the ocean during the Late Cretaceous.

2. Does the paper present novel concepts, ideas, tools, or data? This paper doesn't really present any new ideas, but it has many new data and a slightly different approach from previous papers on the same subject. This paper attempts to define the time frame of the Cenomanian-Turonian interval by integrating new radiochronologic data and using more recent astronomical data. Cyclostratigraphic analysis is performed on data in part different than previously.

3. Are substantial conclusions reached? No, because this article does not stand out enough from that of Mitchell et al., 2008 and the differences in interpretation are not

sufficiently justified.

4. Are the scientific methods and assumptions valid and clearly outlined? More or less
The assumptions seem to be more or less valid, but there are too many assumptions
For example: - The correlation between MS et chert ; - The link between the different
proxies studied and the carbon cycle - The contribution of nutrients from Caribbean
plateau activity. One may ask how the transfer of material in view of the cenoma-
nian paleogeographic configuration is. I think, as authors, both the climate and the
Caribbean plateau activity are at the origin of the Cenomanian-Turonian anoxic never-
theless this paper does not really show it.

5. Are the results sufficient to support the interpretations and conclusions? No 1) Be-
cause there is no discussion on the choice and the climatic significance of the different
proxies studied. Why do studied proxies differ according to stratigraphic interval? Un-
fortunately these proxies do not have the same meaning: The reflectance is controlled
by the lithology. The SiO2 concentration is function of both detrital influx variations
and authigenic / biogenic silica content. The concentration of Al2O3 and TiO2 reflects
changes in detrital flow. Magnetic Susceptibility (MS) variations are function of the
concentration in dia - para and ferromagnetic minerals. How do you explain the in-
crease in MS in levels rich in diamagnetic minerals? Is it strange? Have you done a
statistical analysis which shows the correlation between MS and authigenic/ biogenic
SiO2 content? 2) How did you measure the $\delta$13 C in chert? These analyzes do not
explain what is the minimum carbonate content for valid $\delta$13 C values? 3) The au-
thors state Âń we procure insights in the relationship between orbital forcing and the
Late Cretaceous carbone cycle by deciphering the imprint of astronomical cycles on
lithologic, geophysical and stable isotope records. . . Âż but the data shows that the
imprint of astronomical cycles in the stable isotope records and specially $\delta$13C is very
difficult for deciphering, that's why, the cyclostratigraphic analysis is applied to others
proxies whose link with the carbon cycle is not shown. Some authors' conclusions are
in agreement with Mitchell et al. (2008) works. Mitchell et al. in particular, show a

cyclicity of about 2.4 Ma in the development of anoxia. Unlike Mitchell's works Batenburg et al. suggest that "the exact timing of major carbon cycle perturbations during the Cretaceous may be linked to increased variability in seasonality partner after the prolonged avoidance of seasonal extreme" at the 2.4 Myr scale. This interpretation is not confirmed on any figure. We don't see the 2.4 Myr cycles on Figure 3. Why are not the insolation variations calculated from La2011 data presented?

5. Is the description of experiments and calculations sufficiently complete and precise to allow their reproduction by fellow scientists (traceability of results)? Yes, but scientific reasoning should be more explicit

6. Do the authors give proper credit to related work and clearly indicate their own new/original contribution? I don't doubt the quality of the data, but the choice of these data should be better explained. Their own new contribution is clearly indicated.

7. Does the title clearly reflect the contents of the paper? With this title and content, this article does not stand out enough of Mitchell et al. (2008) works.

8. Does the abstract provide a concise and complete summary? Yes

9. Is the overall presentation well structured and clear? I think the section "results" requires a total reorganization. Before addressing the proxy data and the link with the lithology, we should discuss the time frame of these series (radioisotopic dating + correlation). Any cyclostratigraphic analysis must begin with an accurate (bio)chronological framework. The authors indicate, correctly, that the stratigraphic timing is not based on biostratigraphic, but chemostratigraphic correlations with well-dated series. I believe in the validity of such correlation, but nevertheless to valid a correlation, two continuous chemostratigraphic records must be correlated, which is not the case in this work (see Figure 9). Figure 9 is not convincing and not valid since it lacks isotopic data of the Bonarelli level. On the other hand, this figure is misplaced. It should be positioned at the beginning of the article. Thus, a part of the results and some figures should be reorganized. Another figure that shows the link between $\delta$ 13C and 2.4 kyr orbital

cyclicity should be integrated.

10. Is the language fluent and precise? Yes

11. Are mathematical formulae, symbols, abbreviations, and units correctly defined and used? Yes

12. Should any parts of the paper (text, formulae, figures, tables) be clarified, reduced, combined, or eliminated? Yes, In "Geological setting and proxy records" paragraph, the choice of proxies studied and their meanings must be explained. The "result" paragraph must be reorganized. Correlations and 2.4 kyr orbital cyclicity must be better argued. The modified Figure 9 should be placed at the beginning of the Article. The synthetic Figure 2 should be placed at end of the article.

13. Are the number and quality of references appropriate? Yes, but it is necessary to include additional references to explain the significance of the studied proxies

14. Is the amount and quality of supplementary material appropriate? There are not any

---

## Short Comment (SC1) · 28 Feb 2016

Dear authors,

congratulations on excellent data and an interesting paper. This study is an important contribution, although it might benefit from a better explanation of your approach to astronomical tuning. Could you please comment on the following points?

Published studies (Mitchell et al. 2008; Lanci et al. 2010) suggest relatively uniform sedimentation rates throughout the Furlo section (except of the Bonarelli L.). Your tuning options 1 and 2 imply markedly increased sedimentation rates (or reduced compaction) in the uppermost $\sim$3 m beneath the Bonarelli Level (from $\sim$1 cm/kyr to approximately 1.5 cm/kyr) and results in a $\sim$100 kyr difference relative to the published age models. I realize that this part of the Furlo section is particularly difficult to interpret. Your L* data look great, and after examining your figures in detail I believe your age model might be correct (the apparent increase in both spacing and thicknesses of organic-rich beds in this interval are consistent with your interpretation). As it is, however, your tuning in this interval does not look very convincing. In section 3.3, lines 20-21 you explain that the identification of 405-kyr maxima and minima is based on a 3-5 m bandpass of L* data at Furlo. In both tuning options, however, the uppermost bandpassed maximum below the Bonarelli Level is out-of-phase relative to the 405-kyr maximum in La2011 to which it is correlated. You are apparently using other criteria, but they are not explained. I assume the correlation is based on the bundling of organic-rich beds. This aspect is, however, also problematic, because your lithological log for this interval shows important differences from L*, and it is not clear which of these two is used to define the bundles. For example, the circumflex that should mark the uppermost organic-rich bundle beneath the Bonarelli Level is centered at an exceptionally thick limestone in the lithological log (Fig. 3); this seems to contradict the definition of organic-rich bundles. It would be very helpful if you could show the detail of this part of the section and comment on the differences between your lithological log and color reflectance data. This is particularly important considering the disagreement between your interpretation and published age models.

Could you please explain why do you prefer tuning option #1 over tuning #2? I believe you have good reasons. Without an explanation (which I cannot find in your manuscript), however, the reader is puzzled especially when considering that your tuning #1 appears incompatible with some of the published radioisotopic/astrochronological estimates for the age of the C/T boundary (cf. Eldrett et al. 2015).

Your argument for a Myr eccentricity node prior to OAE II is based on the observed gap in the black shale occurrence at 483-485 m (page 8, lines 30-31). According to your tuning options, however, this interval experienced a 50-60% increase in sedimentation rates (or decrease in compaction) compared to the rest of the section beneath BL. If

you apply correction for this change in sedimentation rate, then the thickness of the shale-free interval decreases by c. 35 %. Such a correction would make this interval comparable to other 405-kyr minima in this section (e.g., ~471-472 m) and disqualify the argument for a Myr node. The exceptional thickness of dark levels above this interval (page 8, lines 31-32) can be attributed to the overall increase in (compacted) sedimentation rates as well.

Recent papers (Jenkyns et al. 2007; Gambacorta et al. 2015) reinterpreted the timing of Bonarelli Level at Furlo and Bottaccione relative to the phases of OAEII. Osmium-isotope excursion marking the onset of the event starts immediately beneath the Bonarelli Level at Furlo (du Vivier et al. 2014). Thus, the possibility that Bonarelli Level represents only the second buildup phase and plateau (page 7, lines 30-31) seems to be outdated (see, for example, figure 12 in Gambacorta et al. 2015). Does this change affect your estimate of the OAE II duration?

Gambacorta et al. (2015) interpret hiatuses in the upper part of the Bonarelli Level at Furlo and other sites in the Umbria-Marche Basin. Could you please indicate how are these hiatuses considered in your age model?

Let me add a note on the paper by Lanci et al. (2010), which is criticized in your text. The phase calibration in this paper was based on a previous astronomical solution (La2004), and is probably incorrect as you noted. The change of interpretation is, however, not due to an incorrect sampling strategy by Lanci et al. (2010). We recently revisited the topic using the same data and simple numerical models. The results suggest that the omission of precession-paced organic layers in Lanci et al. (2010) does not distort the 100-kyr and 400-kyr eccentricity signatures to a degree that would prevent detection of 405-kyr eccentricity phases (Fig. S1.5 in the supporting information of Laurin et al., in press). I would not say that the sampling in Lanci et al. (2010) was "incorrect" (page 6, line 23 in your paper). It was correct considering that the authors needed to avoid lithological bias to focus on the record of changing bottom-water oxygenation in rock-magnetic properties. They just could not have assessed

precession-scale variability, which is a major advantage of your color reflectance data.

I believe the above issues can be fixed. Your paper includes important data and interpretations, and I am hoping to see the final version published soon.

Yours sincerely, Jiří Laurin (Institute of Geophysics ASCR, Prague; laurin@ig.cas.cz)

REFERENCES

Du Vivier ADC, Selby D, Sageman BB, Jarvis I, GroÌĹcke DR, Voigt S (2014) Marine 187Os/188Os isotope stratigraphy reveals the interaction of volcanism and ocean circulation during Oceanic Anoxic Event 2. Earth Planet. Sci. Lett. 389: 23-33, doi:10.1016/j.epsl.2013.12.024.

Eldrett JS, Ma C, Bergman SC, Lutz B, Gregory FJ, Dodsworth P, Phipps M, Hardas P, Minisini D, Ozkan A, Ramezani J, Bowring SA, Kamo SL, Ferguson K, Macaulay C, Kelly AE (2015) An astronomically calibrated stratigraphy of the Cenomanian, Turonian and earliest Coniacian from the Cretaceous Western Interior Seaway, USA: Implications for global chronostratigraphy. Cretaceous Research 56: 316-344, doi: 10.1016/j.cretres.2015.04.010.

Gambacorta G, Jenkyns HC, Russo F, Tsikos H, Wilson PA, Faucher G, Erba E (2015) Carbon- and oxygen-isotope records of mid-Cretaceous Tethyan pelagic sequences from the Umbria–Marche and Belluno Basins (Italy). Newsletters on Stratigraphy 48: 299-323, doi: 10.1127/nos/2015/0066.

Jenkyns HC, Matthews A, Tsikos H, Erel Y (2007) Nitrate reduction, sulfate reduction, and sedimentary iron isotope evolution during the Cenomanian-Turonian oceanic anoxic event. Paleoceanography 22: PA3208, doi:10.1029/2006PA001355.

Lanci L, Muttoni G, Erba E (2010) Astronomical tuning of the Cenomanian Scaglia Bianca Formation at Furlo, Italy. Earth Planet. Sci. Lett. 292: 231-237, doi:10.1016/j.epsl.2010.01.041.

Laurin J, Meyers SR, Galeotti S, Lanci L (in press) Frequency modulation reveals the phasing of orbital eccentricity during Cretaceous Oceanic Anoxic Event II and the Eocene hyperthermals. Earth Planet. Sci. Lett., doi: 10.1016/j.epsl.2016.02.047

Meyers SR, Siewert SE, Singer BS, Sageman BB, Condon DJ, Obradovich JD, Jicha BR, Sawyer DA (2012) Intercalibration of radioisotopic and astrochronologic time scales for the Cenomanian-Turonian boundary interval, Western Interior Basin, USA. Geology 40: 7-10, doi:10.1130/G32261.1.

Mitchell RN, Bice DM, Montanari A, Cleaveland LC, Christianson KT, Coccioni R, Hinnov LA (2008) Oceanic anoxic cycles? Orbital prelude to the Bonarelli Level (OAE 2). Earth Planet. Sci. Lett. 26: 1–16, doi:10.1016/j.epsl.2007.11.026.

———————————————

---

## Referee Comment (RC2) · Anonymous Referee #2 · 5 Apr 2016

This study can be regarded as an extension of an earlier investigation on cyclicity and astrochronology of the same successions, published by Mitchell et al. 2008. This study includes a detailed C-isotope data set and it adds new radioisotope data. The results of this study are mostly in agreement with the earlier study. Additional information is gained on the mode of circulation during OAE2 and on the behaviour of the global carbon cycle before, during and after OAE2.

Carbon isotopes and carbon cycle:

Since the carbon isotope data are the most relevant new data in this study, the carbon isotope results deserve more in-depth discussion. The authors see an obliquity pattern in their data but they do not really discuss these data. In most Cretaceous data sets available from the literature, the obliquity pattern seems not preserved in C-isotope

records, in few others there is some evidence, especially in the amplification of the signal within longer cycles (see Laurin et al.: .... "net transfers between reservoirs are plausibly controlled by ∼1 Myr changes in the amplitude of axial obliquity"). The authors may add some comments on the obliquity – carbon residence time enigma in this study (see also Laurin et al. 2015). They may also discuss possible causes of the remarkable changes in the C-isotope pattern through time. The Turonian C-isotope curve is, across several long eccentricity cycles much more stable than the Cenomanian curve. The authors may also comment on possible reasons, why the C-isotope pattern remains quite noisy throughout two eccentricity cycles from 476m to 484 m.

Climate and oceanography:

It will be important to integrate new information on ocean chemistry, including new Nd-isotope data, into new ocean circulation models. It seems remarkable, that OAE 2 was characterised by a change in Tethys-Atlantic circulation, if Nd-isotope data are integrated into circulation reconstructions (e.g. Martin et al., 2012). An integration of geochemistry into improved circulation models will add value to this study which otherwise may be regarded just as a repetition of the Mitchell et al study. Carbon isotopes and oceanography (p.6): Relatively low values of $\delta$13 C are be associated with stratification of the water column and reduced yearly integrated primary productivity (Sprovieri et al., 2013): » Do these peculiar water mass conditions in the western Tethys control the C-isotope composition of the global marine carbon pool, or do you suggest "global stratification"? Conversely, high $\delta$13 C values likely do reflect good bottom water ventilation during eccentricity minima, with a prolonged avoidance of seasonal extremes, allowing for more stable primary productivity over the annual cycle which may have caused the increase in marine $\delta$13C Âň see e.g. Nd-isotope work by e.g. Martin et al (2012) and others on deep-water formation during OAE 2.

Figures

Please, add a stratigraphy figure to the chapter "geological setting" and to the regional map. This is fundamental information for the reader

---

## Author Comment (AC2) · 16 May 2016

**Author's response: Batenburg et al., 2016**

**J. Laurin**

Dear authors,
Congratulations on excellent data and an interesting paper. This study is an important contribution, although it might benefit from a better explanation of your approach to astronomical tuning. Could you please comment on the following points?
Published studies (Mitchell et al. 2008; Lanci et al. 2010) suggest relatively uniform sedimentation rates throughout the Furlo section (except of the Bonarelli L.). Your tuning options 1 and 2 imply markedly increased sedimentation rates (or reduced compaction) in the uppermost ~3 m beneath the Bonarelli Level (from ~1 cm/kyr to approximately 1.5 cm/kyr) and results in a ~100 kyr difference relative to the published age models. I realize that this part of the Furlo section is particularly difficult to interpret. Your L* data look great, and after examining your figures in detail I believe your age model might be correct (the apparent increase in both spacing and thicknesses of organic-rich beds in this interval are consistent with your interpretation). As it is, however, your tuning in this interval does not look very convincing. In section 3.3, lines 20-21 you explain that the identification of 405-kyr maxima and minima is based on a 3-5 m bandpass of L* data at Furlo. In both tuning options, however, the uppermost bandpassed maximum below the Bonarelli Level is out-of-phase relative to the 405-kyr maximum in La2011 to which it is correlated. You are apparently using other criteria, but they are not explained. I assume the correlation is based on the bundling of organic-rich beds. This aspect is, however, also problematic, because your lithological log for this interval shows important differences from L*, and it is not clear which of these two is used to define the bundles. For example, the circumflex that should mark the uppermost organic-rich bundle beneath the Bonarelli Level is centered at an exceptionally thick limestone in the lithological log (Fig. 3); this seems to contradict the definition of organic-rich bundles. It would be very helpful if you could show the detail of this part of the section and comment on the differences between your lithological log and color reflectance data. This is particularly important considering the disagreement between your interpretation and published age models.

We apologise for the lack of clarity in the tuning of the beds directly underlying the Livello Bonarelli. As this interval is at the edge of the band-pass filter, we prefer to base our tuning on the lithological pattern. The spacing of dark beds is in agreement with the reflectance record, and shows a bundle with thicker cherts and limestones. However, the use of circumflexes to indicate bundles and 100 kyr eccentricity maxima is unclear and introduces some ambiguity. In a revised version of the manuscript, we restrain from using circumflexes. Instead, we use brackets such as "}" spanning whole bundles and their centres as interpreted 100 kyr eccentricity maxima, and we discuss this specific interval in more detail.

Could you please explain why do you prefer tuning option #1 over tuning #2? I believe you have good reasons. Without an explanation (which I cannot find in your manuscript), however, the reader is puzzled especially when considering that your tuning #1 appears incompatible with some of the published radioisotopic/astrochronological estimates for the age of the C/T boundary (cf. Eldrett et al. 2015).

Tuning option #1 is in best agreement with the radioisotopic age for the Mid-Cenomanian event, presented in this study, and the intercalibrated ages for "Ash A" at the base of the *Whiteinella archaeocretacea* zone and the Cenomanian-Turonian boundary. However, tuning #2 is in better agreement with the recently published age of the C/T boundary of Eldrett et al. (2015). Based on our data, we cannot exclude either tuning option, but we discuss the differences with other timescales in more detail.

Your argument for a Myr eccentricity node prior to OAE II is based on the observed gap in the black shale occurrence at 483-485 m (page 8, lines 30-31). According to your tuning options, however, this interval experienced a 50-60% increase in sedimentation rates (or decrease in compaction) compared to the rest of the section beneath BL. If you apply correction for this change in sedimentation rate, then the thickness of the shale-free interval decreases by c. 35 %. Such a correction would make this interval comparable to other 405-kyr minima in this section (e.g., ~471-472 m) and disqualify the argument for a Myr node. The exceptional thickness of dark levels above this interval (page 8, lines 31-32) can be attributed to the overall increase in (compacted) sedimentation rates as well.

When applying such a correction, the gap in black shale occurrence would indeed be similar in thickness to 405 kyr minima lower down in the Furlo section. What makes this interval remarkable, however, is that the number of black levels generally increases upsection, and that more black levels occur per bundle.

Recent papers (Jenkyns et al. 2007; Gambacorta et al. 2015) reinterpreted the timing of Bonarelli Level at Furlo and Bottaccione relative to the phases of OAEII. Osmium-isotope excursion marking the onset of the event starts immediately beneath the Bonarelli Level at Furlo (du Vivier et al. 2014). Thus, the possibility that Bonarelli Level represents only the second buildup phase and plateau (page 7, lines 30-31) seems to be outdated (see, for example, figure 12 in Gambacorta et al. 2015). Does this change affect your estimate of the OAE II duration?

As our age model is not based on chemostratigraphic correlation, our estimate of OAE2 duration is not affected. The duration between the start of the $\delta^{13}$C excursion and the C/T boundary is estimated at 490 kyr. We came to this value by taking into account the estimated duration of the Livello Bonarelli (based on time series analysis of high resolution XRF data), and the 405-kyr tuning for the interval between the top of the Livello Bonarelli and the C/T boundary. We do not consider a large hiatus as proposed by Gambacorta et al (2015) likely, as we have the first occurrence of *Quadrum gartneri* (see also next paragraph). A duration of 490 kyr between the onset of OAE2 and the C/T boundary is, within our stratigraphic uncertainty, in good agreement with the estimate of 538 kyr by Laurin et al. (2016), adapted from the work of Ma et al (2014) and Sageman et al (2006).

Gambacorta et al. (2015) interpret hiatuses in the upper part of the Bonarelli Level at Furlo and other sites in the Umbria-Marche Basin. Could you please indicate how are these hiatuses considered in your age model?

Hiatuses could occur at the sharp shifts in sedimentary facies at the base and the top of the Livello Bonarelli. If such hiatuses are (together) near 405 kyr in duration, they could result in a similar sedimentary rhythm and go unnoticed in our analyses. However, we would expect such a hiatus to have a more pronounced sedimentary expression. The potential for a hiatus at the base of the Livello Bonarelli was previously recognised by Jenkyns et al. (2007), and estimated to be on the order of 20 kyr. Such a hiatus would be small relatively to our tuning target, the 405 kyr periodicity of eccentricity. We would be happy to include a short paragraph on this issue in the revised version of the manuscript.

Let me add a note on the paper by Lanci et al. (2010), which is criticized in your text. The phase calibration in this paper was based on a previous astronomical solution (La2004), and is probably incorrect as you noted. The change of interpretation is, however, not due to an incorrect sampling strategy by Lanci et al. (2010). We recently revisited the topic using the same data and simple numerical models. The results suggest that the omission of precession-paced organic layers in Lanci et al. (2010) does not distort the 100-kyr and 400-kyr eccentricity signatures to a degree that would prevent detection of 405-kyr eccentricity phases (Fig. S1.5 in the supporting information of Laurin et al., in press). I would not say that the sampling in Lanci et al. (2010) was "incorrect" (page 6, line 23 in your paper). It was correct considering that the authors needed to avoid lithological bias to focus on the record of changing bottom-water oxygenation in rock-magnetic properties. They just could not have assessed precession-scale variability, which is a major advantage of your color reflectance data.

We changed the wording of our criticism. However, in the revised version of the manuscript, we are still making clear that, if one does not take into account chert samples, the precession signal is largely eliminated in the intervals of the chert bundles while the weaker precession signal in the limestone beds in between is kept. This is the reason why the precession filtered signal (Fig. 4b in Lanci et al., 2010) suggests the opposite (wrong) phase relation with eccentricity.

I believe the above issues can be fixed. Your paper includes important data and interpretations, and I am hoping to see the final version published soon.
Yours sincerely, Jiˇrí Laurin (Institute of Geophysics ASCR, Prague; laurin@ig.cas.cz)

**References**

Eldrett, J. S., Ma, C., Bergman, S. C., Lutz, B., Gregory, F. J., Dodsworth, P., Phipps, M., Hardas, P., Minisini, D., Ozkan, A., Ramezani, J., Bowring, S. A., Kamo, S. L., Ferguson, K., Macaulay, C., and Kelly, A. E.: An astronomically calibrated stratigraphy of the Cenomanian, Turonian and earliest Coniacian from the Cretaceous Western Interior Seaway, USA: Implications for global chronostratigraphy, Cretaceous Research, 56, 316-344, http://dx.doi.org/10.1016/j.cretres.2015.04.010, 2015.

Gambacorta, G., Jenkyns, H. C., Russo, F., Tsikos, H., Wilson, P. A., Faucher, G., and Erba, E.: Carbon- and oxygen-isotope records of mid-Cretaceous Tethyan pelagic sequences from the Umbria – Marche and Belluno Basins (Italy), Newsletters on Stratigraphy, 48, 299-323, 10.1127/nos/2015/0066, 2015.

Jenkyns, H. C., Matthews, A., Tsikos, H., and Erel, Y.: Nitrate reduction, sulfate reduction, and sedimentary iron isotope evolution during the Cenomanian-Turonian oceanic anoxic event, Paleoceanography, 22, n/a-n/a, 10.1029/2006PA001355, 2007.

Lanci, L., Muttoni, G., and Erba, E.: Astronomical tuning of the Cenomanian Scaglia Bianca Formation at Furlo, Italy, Earth and Planetary Science Letters, 292, 231-237, http://dx.doi.org/10.1016/j.epsl.2010.01.041, 2010.

Laurin, J., Meyers, S. R., Galeotti, S., and Lanci, L.: Frequency modulation reveals the phasing of orbital eccentricity during Cretaceous Oceanic Anoxic Event II and the Eocene hyperthermals, Earth and Planetary Science Letters, 442, 143-156, http://dx.doi.org/10.1016/j.epsl.2016.02.047, 2016.

Ma, C., Meyers, S. R., Sageman, B. B., Singer, B. S., and Jicha, B. R.: Testing the astronomical time scale for oceanic anoxic event 2, and its extension into Cenomanian strata of the Western Interior Basin (USA), Geological Society of America Bulletin, 10.1130/b30922.1, 2014.

Sageman, B. B., Meyers, S. R., and Arthur, M. A.: Orbital time scale and new C-isotope record for Cenomanian-Turonian boundary stratotype, Geology, 34, 125-128, Doi 10.1130/G22074.1, 2006.

---

## Author Comment (AC3) · 16 May 2016

**Author's response: Batenburg et al., 2016**

**Reviewer #2**

This study can be regarded as an extension of an earlier investigation on cyclicity and astrochronology of the same successions, published by Mitchell et al. 2008. This study includes a detailed C-isotope data set and it adds new radioisotope data. The results of this study are mostly in agreement with the earlier study. Additional information is gained on the mode of circulation during OAE2 and on the behaviour of the global carbon cycle before, during and after OAE2.

Carbon isotopes and carbon cycle:
Since the carbon isotope data are the most relevant new data in this study, the carbon isotope results deserve more in-depth discussion. The authors see an obliquity pattern in their data but they do not really discuss these data. In most Cretaceous data sets available from the literature, the obliquity pattern seems not preserved in C-isotope records, in few others there is some evidence, especially in the amplification of the signal within longer cycles (see Laurin et al.: .... "net transfers between reservoirs are plausibly controlled by ~1 Myr changes in the amplitude of axial obliquity"). The authors may add some comments on the obliquity – carbon residence time enigma in this study (see also Laurin et al. 2015).

The potential ~1 Myr cycle in the carbon isotope curve is very interesting and we are happy to include this in the discussion of long term orbital forcing and the carbon cycle in a revised version of the manuscript. In general, it seems that the trends in the carbon cycle follow a ~1 Myr pacing, whereas sharp excursions occur at ~2.4 Myr intervals superimposed on this pattern. The high resolution XRF data from the Livello Bonarelli show a likely obliquity pacing, as detected previously during OAE2 by Meyers et al. (2012). This observation likely reflects a long-term (~2.4 Myr) minimum in eccentricity-modulated precession, during which obliquity can become the dominant astronomical parameter driving changes in insolation, as was observed in the Eocene by Westerhold et al. (2014).

They may also discuss possible causes of the remarkable changes in the C-isotope pattern through time. The Turonian C- isotope curve is, across several long eccentricity cycles much more stable than the Cenomanian curve.
The authors may also comment on possible reasons, why the C- isotope pattern remains quite noisy throughout two eccentricity cycles from 476m to 484 m.

The Cenomanian carbon isotope curve indeed seems much more strongly paced by the 405 kyr cycle of eccentricity modulated precession than the Turonian $\delta^{13}$C curve. A similar phenomenon was observed for the end of the Albian, and interpreted to reflect a change to a more stable ocean circulation pattern (Giorgioni et al., 2012). For the Cenomanian/Turonian, the carbon cycle may have become more stable as $CO_2$ may have been drawn down by black shale formation and volcanic activity, delivering nutrients, may have decreased.
The high variability in $\delta^{13}$C values from 476 to 484 m coincides with the frequent occurrence of organic-matter rich intervals. Potentially, this may have influenced the $\delta^{13}$C values of, for example, early diagenetic cements.

Climate and oceanography:
It will be important to integrate new information on ocean chemistry, including new Nd- isotope data, into new ocean circulation models. It seems remarkable, that OAE 2 was characterised by a change in Tethys-Atlantic circulation, if Nd-isotope data are integrated into circulation reconstructions (e.g. Martin et al., 2012). An integration of geochemistry into improved circulation models will add value to this study which otherwise may be regarded just as a repetition of the Mitchell et al study. Carbon isotopes and oceanography (p.6): Relatively low values of δ13C are be associated with stratification of the water column and reduced yearly integrated primary productivity (Sprovieri et al., 2013): » Do these peculiar water mass conditions in the western Tethys control the C-isotope composition of the global marine carbon pool, or do you suggest "global stratification"? Conversely, high δ13C values likely do reflect good bottom water ventilation during eccentricity minima, with a prolonged avoidance of seasonal extremes, allowing for more stable primary productivity over the annual cycle which may have caused the increase

in marine δ13C Ăˇn see e.g. Nd-isotope work by e.g. Martin et al (2012) and others on deep-water formation during OAE 2.

We agree with the reviewer that new Nd-isotope data are essential for a better understanding of ocean circulation before and during OAE2. Although beyond the scope of this manuscript, this is part of the ongoing research by S. J. Batenburg. The Nd-isotope of Martin et al (2012) and others indicates a change in deep water formation and exchange at the Cenomanian-Turonian transition, which may have driven the transport of nutrients and black shale deposition. Such a change in circulation may have resulted from astronomically-forced changes in seasonality and the hydrological cycle.

As mentioned in the reply to reviewer #1, we highlight the differences between our manuscript and the Mitchell et al. (2008) study in the revised version of the manuscript. The remarks on the patterns in $\delta^{13}$C values refer to the western Tethys, where regional anoxia already developed episodically before OAE2.

Figures
Please, add a stratigraphy figure to the chapter "geological setting" and to the regional map. This is fundamental information for the reader.

We gladly include a generalised stratigraphy of the Umbria-Marche basin next to the regional map in a revised version of the manuscript.

**References**

Giorgioni, M., Weissert, H., Bernasconi, S. M., Hochuli, P. A., Coccioni, R., and Keller, C. E.: Orbital control on carbon cycle and oceanography in the mid-Cretaceous greenhouse, Paleoceanography, 27, 10.1029/2011PA002163, 2012.

Martin, E. E., MacLeod, K. G., Jiménez Berrocoso, A., and Bourbon, E.: Water mass circulation on Demerara Rise during the Late Cretaceous based on Nd isotopes, Earth and Planetary Science Letters, 327–328, 111-120, http://dx.doi.org/10.1016/j.epsl.2012.01.037, 2012.

Meyers, S. R., Sageman, B. B., and Arthur, M. A.: Obliquity forcing of organic matter accumulation during Oceanic Anoxic Event 2, Paleoceanography, 27, 10.1029/2012pa002286, 2012.

Mitchell, R. N., Bice, D. M., Montanari, A., Cleaveland, L. C., Christianson, K. T., Coccioni, R., and Hinnov, L. A.: Oceanic anoxic cycles? Orbital prelude to the Bonarelli Level (OAE 2), Earth and Planetary Science Letters, 267, 1-16, DOI 10.1016/j.epsl.2007.11.026, 2008.

Westerhold, T., Röhl, U., Pälike, H., Wilkens, R., Wilson, P. A., and Acton, G.: Orbitally tuned timescale and astronomical forcing in the middle Eocene to early Oligocene, Clim. Past, 10, 955-973, 10.5194/cp-10-955-2014, 2014.

---

## Author Response (AR1)

**Authors' response: Batenburg et al., 2016**

**Detailed response to the reviews**

The reviewers' comments are copied in grey, our original response in black, and the applied changes are indicated in blue. Page and line numbers in blue refer to the revised manuscript. The replies are followed by a list of relevant changes and a marked-up version of the manuscript.

**Reviewer #1**

1. Does the paper address relevant scientific questions within the scope of CP? Yes, the authors want to show and explain climate control on the development of poor oxygenation conditions in the ocean during the Late Cretaceous.
2. Does the paper present novel concepts, ideas, tools, or data? This paper doesn't really present any new ideas, but it has many new data and a slightly different approach from previous papers on the same subject. This paper attempts to define the time frame of the Cenomanian-Turonian interval by integrating new radiochronologic data and using more recent astronomical data. Cyclostratigraphic analysis is performed on data in part different than previously.
3. Are substantial conclusions reached? No, because this article does not stand out enough from that of Mitchell et al., 2008 and the differences in interpretation are not sufficiently justified.

We are happy to read that the reviewer recognizes the significant amount of new paleoclimate proxy data that is presented in our manuscript, as well as a new radioisotopic date for the Cenomanian. Still, the reviewer has the opinion that that our manuscript does not stand out enough from Mitchell et al. (2008). In the original version of the manuscript, we have listed a number of differences in tuning approach and interpretation between Mitchell et al. (2008) and our manuscript. One important difference is that we solely use the stable 405 kyr periodicity of eccentricity as a tuning target, whereas Mitchell et al. (2008) also tuned to 100-kyr eccentricity. We adopt this tuning strategy because the 405-kyr component is the only astronomical tuning target that can be used beyond ~50 Ma, and is thus the prime target in the Mesozoic. The C/T boundary age might seem similar between the two papers, but this is in fact not the case. Mitchell et al. (2008) used radioisotopic ages of Sageman et al. (2006), who used the Fish Canyon sanidine standard age of 28.02 ± 0.28 Ma of Renne et al. (1998), while we use the 28.201 ± 0.046 Ma of Kuiper et al. (2008). At around 94 Ma, this makes a difference of more than 600 kyr or 1.5 x 405-kyr cycle. This improvement is critical for extending the astronomical time scale from the K/Pg boundary back to the C/T boundary and beyond.

The discussion of differences in approach between Mitchell et al. (2008) and our paper, however, did not seem to be effective in indicating the fundamental differences that exists between both works. Therefore, we recognize that this part of the manuscript should be improved. In the revised version of the manuscript, we are paying special attention to the discussion of the differences in cyclostratigraphic approach and paleoenvironmental interpretation between Mitchell et al. (2008) and Batenburg et al. (2016). This involves a fundamental rewriting of the paragraphs involved.

We include a new paragraph on page 11 (l.10–17) summarising the new aspects of our study, including 1) the use of only the 405-kyr periodicity component of eccentricity from the new La2011 solution, 2) the independent estimate of the duration of deposition of the Livello Bonarelli, 3) the consequences of the use of the new intercalibrated age of the Fish Canyon sanidine and 4) the new radioisotopic age of the mid-Cenomanian event. Points 1) and 3) are particularly relevant in comparison to the Mitchell et al. (2008) study, and are discussed further on page 11, l. 18–25 and page 13, l. 4–11, respectively. In addition, on page 16, lines 16–21 explain the difference in invoked climatic mechanisms between Mitchell et al. (2008) and the present study.

4. Are the scientific methods and assumptions valid and clearly outlined? More or less The assumptions seem to be more or less valid, but there are too many assumptions. For example: - The correlation between MS et chert ; - The link between the different proxies studied and the carbon cycle - The contribution of nutrients from Caribbean plateau activity. One may ask how the transfer of material in view of the cenomanian paleogeographic configuration is. I think, as

authors, both the climate and the Caribbean plateau activity are at the origin of the Cenomanian-Turonian anoxic nevertheless this paper does not really show it.

Figure 3 shows the relationship between MS and cherts, as well as the relation with different proxies. The link between this study and the long-term behavior of the carbon cycle is discussed in paragraph 4.4, as is the likely supply of nutrients from the Caribbean LIP. Volcanism was probably the ultimate driver of oceanic anoxia, but, based on our findings, astronomical forcing likely determined the exact timing. We focus on the astronomical forcing aspect and consider the precise oceanographic processes and geochemical pathways beyond the scope of this paper.

In section 4.3.1 (p.14) on the expression of long-term eccentricity forcing, we explain the patterns observed in the lithology and the carbon isotope data that are indicative of the behaviour of the 2.4 Myr cycle of eccentricity in more detail. We have also revised the following section on the relation with volcanism (4.3.2, p. 15-16) to include more specific information on previously published studies on volcanic activity and ocean circulation during the Cenomanian–Turonian transition to put our findings in a larger context.

5. Are the results sufficient to support the interpretations and conclusions? No
5.1 Because there is no discussion on the choice and the climatic significance of the different proxies studied. Why do studied proxies differ according to stratigraphic interval? Unfortunately these proxies do not have the same meaning: The reflectance is controlled by the lithology. The SiO2 concentration is function of both detrital influx variations and authigenic / biogenic silica content. The concentration of Al2O3 and TiO2 reflects changes in detrital flow. Magnetic Susceptibility (MS) variations are function of the concentration in dia-, para and ferromagnetic minerals. How do you explain the increase in MS in levels rich in diamagnetic minerals? Is it strange? Have you done a statistical analysis which shows the correlation between MS and authigenic/ biogenic SiO2 content?

Different paleo-environmental proxies have been measured between the Scaglia Rossa/Bianca Formations on the one hand (C and O stable isotopes, magnetic susceptibility, reflectivity and limestone-chert alternations), and the Livello Bonarelli on the other hand (XRF-derived $SiO_2$, $Al_2O_3$, $TiO_2$ and Loss-on-Ignition). The large differences in sedimentary facies imply that different proxy records form the best archives for paleoclimatic variability. For example, reflectance data are very useful in the interval where black shales occur, whereas colour variations in the Scaglia Bianca (above the Livello Bonarelli) are limited. We discuss the characteristics of the different proxies in terms of paleoclimatic and paleo-environmental interpretation in more detail in the revised version of the manuscript. As the individual proxy records are limited to the lithological units, we do not have overlapping magnetic susceptibility and $SiO_2$ data, and cannot perform statistical analyses.

The choice and significance of the proxies is now discussed more thoroughly in section 2.1 (Geological setting and proxy records), in particular on page 4, lines 13–25, and in section 4.1 (Proxy records and correlation of the C/T boundary) from page 8, line 21 to page 9, line 9.

5.2 How did you measure the δ13 C in chert? These analyzes do not explain what is the minimum carbonate content for valid δ13 C values?

$δ^{13}C$ values were obtained from powdered samples, of the limestones and marls as well as of the cherts, which still contained carbonate. If, in the first measurement run, carbonate concentrations were too low to obtain a reliable signal, measurements were repeated with a larger volume. This information is being incorporated in the revised manuscript.

This information is inserted on page 5, lines 3–4.

5.3 The authors state "we procure insights in the relationship between orbital forcing and the ´ Late Cretaceous carbone cycle by deciphering the imprint of astronomical cycles on lithologic, geophysical and stable isotope records. . . " but the data shows that the imprint of astronomical cycles in the stable isotope records and specially δ13C is very difficult for deciphering, that's why, the cyclostratigraphic analysis is applied to others proxies whose link with the carbon cycle is not shown.

The cyclostratigraphic framework is indeed constructed based primarily on the geophysical proxies and the limestone-chert alternations. In Figure 2, we show that eccentricity maxima, as interpreted from these proxies, correspond to high variability in $\delta^{13}C$, as well as with a tendency towards more negative $\delta^{13}C$ values. This is one example of how we link the cyclostratigraphic interpretations with the global carbon cycle. Additionally, one of the significant findings reported in this manuscript, is the fact that the base of the Livello Bonarelli corresponds to the first 100-kyr eccentricity maximum after a 405-kyr eccentricity minimum, which is another clear example of a link between cyclostratigraphy and global carbon cycle perturbations.

The construction of our cyclostratigraphic framework is discussed in more detail in section 4.2.2 (Calibration to 405-kyr eccentricity), on page 11, lines 4–25.

> Some authors' conclusions are in agreement with Mitchell et al. (2008) works. Mitchell et al. in particular, show a cyclicity of about 2.4 Ma in the development of anoxia. Unlike Mitchell's works Batenburg et al. suggest that "the exact timing of major carbon cycle perturbations during the Cretaceous may be linked to increased variability in seasonality partner after the prolonged avoidance of seasonal extreme" at the 2.4 Myr scale. This interpretation is not confirmed on any figure. We don't see the 2.4 Myr cycles on Figure 3.

A possible role of the 2.4 Myr eccentricity cycle is discussed after the observation that the mid-Cenomanian event, the OAE-2 and the Pewsey excursion are separated by 2.0 - 2.4 Myr respectively, and the observation that black shales are lacking in the interval preceding OAE2. This lithological pattern and the spacing between the three "events" can be observed in Figure 3.

Sections 4.3.1 and 4.3.2 (pages 14–16) discuss the observations on the influence of the 2.4 Myr periodicity of eccentricity in more detail, as well as the effects of the suggested particular orbital configuration on climate and oceanography.

> Why are not the insolation variations calculated from La2011 data presented?

This is because the most recent insolation solution that is currently available is La2004. This insolation solution is only valid until ~40 Ma. The La2011 and La2010 solutions that are available at present, are eccentricity-only solutions. Only the 405-kyr eccentricity component of the La2010 and La2011 solutions can be used in the Cenomanian/Turonian interval.

6. Is the description of experiments and calculations sufficiently complete and precise to allow their reproduction by fellow scientists (traceability of results)? Yes, but scientific reasoning should be more explicit

We agree with the reviewer that scientific reasoning should be more straight-forward and explicit in the revised version of the manuscript. Therewith, we first and foremost focus on [1] showing the differences with the Mitchell et al. (2008) approach and [2] the hypothesis that OAE2 was favored by a specific sequence of astronomical configurations, with a prolonged period of low eccentricity (2.4 Myr eccentricity minimum) followed by an eccentricity maximum (100-kyr eccentricity maximum).

We have thoroughly revised the Discussion sections (see comments in blue above).

7. Do the authors give proper credit to related work and clearly indicate their own new/original contribution? I don't doubt the quality of the data, but the choice of these data should be better explained. Their own new contribution is clearly indicated.
8.
9. Does the title clearly reflect the contents of the paper? With this title and content, this article does not stand out enough of Mitchell et al. (2008) works.

See above.

10. Does the abstract provide a concise and complete summary? Yes
11. Is the overall presentation well structured and clear? I think the section "results" requires a total reorganization. Before addressing the proxy data and the link with the lithology, we should discuss the time frame of these series (radioisotopic dating + correlation). Any cyclostratigraphic

analysis must begin with an accurate (bio)chronological framework. The authors indicate, correctly, that the stratigraphic timing is not based on biostratigraphic, but chemostratigraphic correlations with well-dated series. I believe in the validity of such correlation, but nevertheless to valid a correlation, two continuous chemostratigraphic records must be correlated, which is not the case in this work (see Figure 9). Figure 9 is not convincing and not valid since it lacks isotopic data of the Bonarelli level. On the other hand, this figure is misplaced. It should be positioned at the beginning of the article. Thus, a part of the results and some figures should be reorganized. Another figure that shows the link between δ 13C and 2.4 kyr orbital cyclicity should be integrated.

We do not agree with the reviewer that our manuscript needs a total reorganization. The reviewer rightfully says that *"any cyclostratigraphic analysis must begin with an accurate (bio)chronological framework"*. It goes without saying that we agree with this statement. Hence, the well-studied and well-documented biostratigraphic framework is presented early in the manuscript, in Figure 3, alongside the lithological column. Unfortunately, the reviewer misinterpreted our Figure 9, as this figure is not meant to constrain the initial stratigraphic framework by correlating outstanding features in d13C. Instead, Figure 9 shows a chemostratigraphic comparison between carbon isotope records from contemporaneous sections. In this figure, we show carbon isotope records along their original age-models, as constructed by the authors of the publications in which these data have been presented. In other words, Figure 9 is a figure in which we evaluate our tuning, and therefore, it should be presented after we presented the two tuning options for the studied sections. It is likely that the confusion was caused by the early call-out to Figure 9 in the original manuscript (Section 4.1), before the actual tuning is presented. In the revised version of the manuscript we discuss Figure 9 _after_ we presented our tuning options for the studied sections.

The independent time constraints are now discussed more explicitly in section 4.2.3 (Integration with radioisotopic ages) in addition to the correlation of the C/T boundary (page 9, lines 10–22).

12. Is the language fluent and precise? Yes
13. Are mathematical formulae, symbols, abbreviations, and units correctly defined and used? Yes
14. Should any parts of the paper (text, formulae, figures, tables) be clarified, reduced, combined, or eliminated? Yes, In "Geological setting and proxy records" paragraph, the choice of proxies studied and their meanings must be explained. The "result" paragraph must be reorganized. Correlations and 2.4 kyr orbital cyclicity must be better argued. The modified Figure 9 should be placed at the beginning of the Article. The synthetic Figure 2 should be placed at end of the article.

These comments have all been addressed above.

15. Are the number and quality of references appropriate? Yes, but it is necessary to include additional references to explain the significance of the studied proxies
16. Is the amount and quality of supplementary material appropriate? There are not any

**J. Laurin**

Dear authors,
Congratulations on excellent data and an interesting paper. This study is an important contribution, although it might benefit from a better explanation of your approach to astronomical tuning. Could you please comment on the following points?

Published studies (Mitchell et al. 2008; Lanci et al. 2010) suggest relatively uniform sedimentation rates throughout the Furlo section (except of the Bonarelli L.). Your tuning options 1 and 2 imply markedly increased sedimentation rates (or reduced compaction) in the uppermost ~3 m beneath the Bonarelli Level (from ~1 cm/kyr to approximately 1.5 cm/kyr) and results in a ~100 kyr difference relative to the published age models. I realize that this part of the Furlo section is particularly difficult to interpret. Your L* data look great, and after examining your figures in detail I believe your age model might be correct (the apparent increase in both spacing and thicknesses of organic-rich beds in this interval are consistent with your interpretation). As it is, however, your tuning in this interval does not look very convincing. In section 3.3, lines 20-21 you explain that the identification of 405-kyr maxima and minima is based on a 3-5 m bandpass of L* data at Furlo. In both tuning options, however, the uppermost bandpassed maximum below the Bonarelli Level is out-of-phase relative to the 405-kyr maximum in La2011 to which it is correlated. You are apparently using other criteria, but they are not explained. I assume the correlation is based on the bundling of organic-rich beds. This aspect is, however, also problematic, because your lithological log for this interval shows important differences from L*, and it is not clear which of these two is used to define the bundles. For example, the circumflex that should mark the uppermost organic-rich bundle beneath the Bonarelli Level is centered at an exceptionally thick limestone in the lithological log (Fig. 3); this seems to contradict the definition of organic-rich bundles. It would be very helpful if you could show the detail of this part of the section and comment on the differences between your lithological log and color reflectance data. This is particularly important considering the disagreement between your interpretation and published age models.

We apologise for the lack of clarity in the tuning of the beds directly underlying the Livello Bonarelli. As this interval is at the edge of the band-pass filter, we prefer to base our tuning on the lithological pattern. The spacing of dark beds is in agreement with the reflectance record, and shows a bundle with thicker cherts and limestones. However, the use of circumflexes to indicate bundles and 100 kyr eccentricity maxima is unclear and introduces some ambiguity. In a revised version of the manuscript, we restrain from using circumflexes. Instead, we use brackets such as "}" spanning whole bundles and their centres as interpreted 100 kyr eccentricity maxima, and we discuss this specific interval in more detail.

Figure 2 has been adjusted, and the interval below the Livello Bonarelli is discussed on page 6, lines 18–22, as well as on page 14, lines 13–21.

Could you please explain why do you prefer tuning option #1 over tuning #2? I believe you have good reasons. Without an explanation (which I cannot find in your manuscript), however, the reader is puzzled especially when considering that your tuning #1 appears incompatible with some of the published radioisotopic/astrochronological estimates for the age of the C/T boundary (cf. Eldrett et al. 2015).

Tuning option #1 is in best agreement with the radioisotopic age for the Mid-Cenomanian event, presented in this study, and the intercalibrated ages for "Ash A" at the base of the *Whiteinella archaeocretacea* zone and the Cenomanian-Turonian boundary. However, tuning #2 is in better agreement with the recently published age of the C/T boundary of Eldrett et al. (2015). Based on our data, we cannot exclude either tuning option, but we discuss the differences with other timescales in more detail.

We have included a paragraph (page 13, lines 25–30) comparing the tuning options to radioisotopic ages and previous studies.

Your argument for a Myr eccentricity node prior to OAE II is based on the observed gap in the black shale occurrence at 483-485 m (page 8, lines 30-31). According to your tuning options, however, this interval experienced a 50-60% increase in sedimentation rates (or decrease in compaction) compared to the rest of the section beneath BL. If you apply correction for this change in sedimentation rate,

then the thickness of the shale-free interval decreases by c. 35 %. Such a correction would make this interval comparable to other 405-kyr minima in this section (e.g., ~471-472 m) and disqualify the argument for a Myr node. The exceptional thickness of dark levels above this interval (page 8, lines 31-32) can be attributed to the overall increase in (compacted) sedimentation rates as well.

When applying such a correction, the gap in black shale occurrence would indeed be similar in thickness to 405 kyr minima lower down in the Furlo section. What makes this interval remarkable, however, is that the number of black levels generally increases upsection, and that more black levels occur per bundle.

We discuss our observation on the interval below the Livello Bonarelli in more detail on page 14, lines 13–21, and include more information on other observations on the expression of 2.4 Myr eccentricity forcing in section 4.3.1 (p. 14-15).

Recent papers (Jenkyns et al. 2007; Gambacorta et al. 2015) reinterpreted the timing of Bonarelli Level at Furlo and Bottaccione relative to the phases of OAEII. Osmium-isotope excursion marking the onset of the event starts immediately beneath the Bonarelli Level at Furlo (du Vivier et al. 2014). Thus, the possibility that Bonarelli Level represents only the second buildup phase and plateau (page 7, lines 30-31) seems to be outdated (see, for example, figure 12 in Gambacorta et al. 2015). Does this change affect your estimate of the OAE II duration?

As our age model is not based on chemostratigraphic correlation, our estimate of OAE2 duration is not affected. The duration between the start of the $\delta^{13}$C excursion and the C/T boundary is estimated at 490 kyr. We came to this value by taking into account the estimated duration of the Livello Bonarelli (based on time series analysis of high resolution XRF data), and the 405-kyr tuning for the interval between the top of the Livello Bonarelli and the C/T boundary. We do not consider a large hiatus as proposed by Gambacorta et al (2015) likely, as we have the first occurrence of *Quadrum gartneri* (see also next paragraph). A duration of 490 kyr between the onset of OAE2 and the C/T boundary is, within our stratigraphic uncertainty, in good agreement with the estimate of 538 kyr by Laurin et al. (2016), adapted from the work of Ma et al (2014) and Sageman et al (2006).

Gambacorta et al. (2015) interpret hiatuses in the upper part of the Bonarelli Level at Furlo and other sites in the Umbria-Marche Basin. Could you please indicate how are these hiatuses considered in your age model?

Hiatuses could occur at the sharp shifts in sedimentary facies at the base and the top of the Livello Bonarelli. If such hiatuses are (together) near 405 kyr in duration, they could result in a similar sedimentary rhythm and go unnoticed in our analyses. However, we would expect such a hiatus to have a more pronounced sedimentary expression. The potential for a hiatus at the base of the Livello Bonarelli was previously recognised by Jenkyns et al. (2007), and estimated to be on the order of 20 kyr. Such a hiatus would be small relatively to our tuning target, the 405 kyr periodicity of eccentricity. We would be happy to include a short paragraph on this issue in the revised version of the manuscript.

The discussion on the two points above is now included in the manuscript on page 12, lines 20-29.

Let me add a note on the paper by Lanci et al. (2010), which is criticized in your text. The phase calibration in this paper was based on a previous astronomical solution (La2004), and is probably incorrect as you noted. The change of interpretation is, however, not due to an incorrect sampling strategy by Lanci et al. (2010). We recently revisited the topic using the same data and simple numerical models. The results suggest that the omission of precession-paced organic layers in Lanci et al. (2010) does not distort the 100-kyr and 400-kyr eccentricity signatures to a degree that would prevent detection of 405-kyr eccentricity phases (Fig. S1.5 in the supporting information of Laurin et al., in press). I would not say that the sampling in Lanci et al. (2010) was "incorrect" (page 6, line 23 in your paper). It was correct considering that the authors needed to avoid lithological bias to focus on the record of changing bottom-water oxygenation in rock-magnetic properties. They just could not have assessed precession-scale variability, which is a major advantage of your color reflectance data.

We changed the wording of our criticism. However, in the revised version of the manuscript, we are still making clear that, if one does not take into account chert samples, the precession signal is largely eliminated in the intervals of the chert bundles while the weaker precession signal in the limestone beds

in between is kept. This is the reason why the precession filtered signal (Fig. 4b in Lanci et al., 2010) suggests the opposite (wrong) phase relation with eccentricity.

The wording on page 10, lines 9–14 is revised and includes a reference to the new interpretation of the data of Lanci et al. (2010) by Laurin et al (2016).

I believe the above issues can be fixed. Your paper includes important data and interpretations, and I am hoping to see the final version published soon.
Yours sincerely, Jiˇrí Laurin (Institute of Geophysics ASCR, Prague; laurin@ig.cas.cz)

**Anonymous Referee #2**

This study can be regarded as an extension of an earlier investigation on cyclicity and astrochronology of the same successions, published by Mitchell et al. 2008. This study includes a detailed C-isotope data set and it adds new radioisotope data. The results of this study are mostly in agreement with the earlier study. Additional information is gained on the mode of circulation during OAE2 and on the behaviour of the global carbon cycle before, during and after OAE2.

Carbon isotopes and carbon cycle:
Since the carbon isotope data are the most relevant new data in this study, the carbon isotope results deserve more in-depth discussion. The authors see an obliquity pattern in their data but they do not really discuss these data. In most Cretaceous data sets available from the literature, the obliquity pattern seems not preserved in C-isotope records, in few others there is some evidence, especially in the amplification of the signal within longer cycles (see Laurin et al.: .... "net transfers between reservoirs are plausibly controlled by ~1 Myr changes in the amplitude of axial obliquity"). The authors may add some comments on the obliquity – carbon residence time enigma in this study (see also Laurin et al. 2015).

The potential ~1 Myr cycle in the carbon isotope curve is very interesting and we are happy to include this in the discussion of long term orbital forcing and the carbon cycle in a revised version of the manuscript. In general, it seems that the trends in the carbon cycle follow a ~1 Myr pacing, whereas sharp excursions occur at ~2.4 Myr intervals superimposed on this pattern. The high resolution XRF data from the Livello Bonarelli show a likely obliquity pacing, as detected previously during OAE2 by Meyers et al. (2012). This observation likely reflects a long-term (~2.4 Myr) minimum in eccentricity-modulated precession, during which obliquity can become the dominant astronomical parameter driving changes in insolation, as was observed in the Eocene by Westerhold et al. (2014).

These observations are discussed in a new paragraph within section 4.3.1, on page 15, lines 8-24.

They may also discuss possible causes of the remarkable changes in the C-isotope pattern through time. The Turonian C- isotope curve is, across several long eccentricity cycles much more stable than the Cenomanian curve.
The authors may also comment on possible reasons, why the C- isotope pattern remains quite noisy throughout two eccentricity cycles from 476m to 484 m.

The Cenomanian carbon isotope curve indeed seems much more strongly paced by the 405 kyr cycle of eccentricity modulated precession than the Turonian $\delta^{13}$C curve. A similar phenomenon was observed for the end of the Albian, and interpreted to reflect a change to a more stable ocean circulation pattern (Giorgioni et al., 2012). For the Cenomanian/Turonian, the carbon cycle may have become more stable as $CO_2$ may have been drawn down by black shale formation and volcanic activity, delivering nutrients, may have decreased.

This view on the stability of the Turonian carbon isotope curve with respect to the Cenomanian curve is included on page 15, lines 25-29.

The high variability in $\delta^{13}$C values from 476 to 484 m coincides with the frequent occurrence of organic-matter rich intervals. Potentially, this may have influenced the $\delta^{13}$C values of, for example, early diagenetic cements.

This information is included on page 8 in lines 27–29.

Climate and oceanography:
It will be important to integrate new information on ocean chemistry, including new Nd- isotope data, into new ocean circulation models. It seems remarkable, that OAE 2 was characterised by a change in Tethys-Atlantic circulation, if Nd-isotope data are integrated into circulation reconstructions (e.g. Martin et al., 2012). An integration of geochemistry into improved circulation models will add value to this study which otherwise may be regarded just as a repetition of the Mitchell et al study. Carbon isotopes and
oceanography (p.6): Relatively low values of δ13C are be associated with stratification of the water column and reduced yearly integrated primary productivity (Sprovieri et al., 2013): » Do these peculiar

water mass conditions in the western Tethys control the C-isotope composition of the global marine carbon pool, or do you suggest "global stratification"? Conversely, high δ13C values likely do reflect good bottom water ventilation during eccentricity minima, with a prolonged avoidance of seasonal extremes, allowing for more stable primary productivity over the annual cycle which may have caused the increase in marine δ13C Aˇn see e.g. Nd-isotope work by e.g. Martin et al (2012) and others on deep-water formation during OAE 2.

We agree with the reviewer that new Nd-isotope data are essential for a better understanding of ocean circulation before and during OAE2. Although beyond the scope of this manuscript, this is part of the ongoing research by S. J. Batenburg. The Nd-isotope of Martin et al (2012) and others indicates a change in deep water formation and exchange at the Cenomanian-Turonian transition, which may have driven the transport of nutrients and black shale deposition. Such a change in circulation may have resulted from astronomically-forced changes in seasonality and the hydrological cycle.
As mentioned in the reply to reviewer #1, we highlight the differences between out manuscript and the Mitchell et al. (2008) study in the revised version of the manuscript. The remarks on the patterns in $\delta^{13}C$ values refer to the western Tethys, where regional anoxia already developed episodically before OAE2.

In the discussion, section 4.3.2 (pages 15-16) has been revised to incorporate more specific information on the potential effect of long-period astronomical forcing on hydrology and circulation. The differences with the Mitchell et al. (2008) study with regard to climatic mechanisms are discussed in more detail in this section, and the differences in tuning approach in section 4.2.2 (see also the reply to reviewer 1).

Figures
Please, add a stratigraphy figure to the chapter "geological setting" and to the regional map. This is fundamental information for the reader.

We gladly include a generalised stratigraphy of the Umbria-Marche basin next to the regional map in a revised version of the manuscript.

A generalised stratigraphy is now incorporated in Figure 1.

Superimposed on the hierarchical stacking patterns of lithologies in the studied succession, several features of the lithological and proxy records reveal the influence of long-term periodicities on local sedimentation and global climate. These observations include: (i) the absence of cherts in an interval below the Livello Bonarelli; (ii) a strong expression of obliquity forcing during deposition of the Livello Bonarelli, contemporaneous with a sedimentary response to the 100-kyr forcing of eccentricity; (iii) a spacing of 2.0 and 2.4 Myr, respectively, between the mid-Cenomanian $\delta^{13}$C excursion, the onset of OAE2, and a positive $\delta^{13}$C excursion in the mid-Turonian. These observations, in combination with a previously noted ~1 Myr cyclicity in $\delta^{13}$C, reveal a pacing of climatic events by long-term eccentricity cycles and will be further discussed in the following paragraphs.
Below the Livello Bonarelli, in the interval 483-485 m, black shales are conspicuously absent. This may partially be due to an increase in sedimentation rate, as indicated by a larger spacing between beds from 483.5 m upwards, but this pattern breaks the trend of an increasing number of black cherts and shales up-section, per meter as well as per interpreted ~100 kyr bundle. This may reflect the prolonged avoidance of seasonal extremes during long-term eccentricity minima of the 2.4-Myr cycle.

**P 15, l 8 – P16, l 15**
Trace element studies indicate that volcanic activity of the Caribbean Large Igneous Province increasingly supplied 
[revised manuscript text omitted]

1.1.1

| Page 11: [3] Deleted | authors | 8/1/16 6:05:00 PM |

**Tuning options**

Because of its stability, only the 405-kyr component of eccentricity can be used for astronomical tuning in the Cretaceous and not the ~100 kyr periodicity that was used in previous tuning efforts (Mitchell et al., 2008). The shorter obliquity and precession terms can only be used for the development of floating time scales.

**Radioisotopic dating**

The new astrochronologies allow for assessing the long-term behavior of the carbon cycle during the C/T transition. The onset of the MCE, the base of the Livello Bonarelli, and the middle of the negative $\delta^{13}C$ excursion of the mid-Turonian are separated by 2.0 Myr and 2.4 Myr, respectively. The 1.6 Myr long negative excursion in the mid-Turonian is characterized by an intermittent double positive peak ("Pewsey events"; Jarvis et al., 2006), similar to the MCE, starting at 91.7 Ma (tuning #1) or 92.1 Ma (tuning #2). These repetitive variations in $\delta^{13}C$, superimposed on the long-term behavior of the carbon cycle, are likely paced by the ~2.4-Myr eccentricity period. Following tuning #1, a tentative comparison with the full eccentricity solution La2011 (Fig. 2) reveals the occurrence of pronounced long-term minima in eccentricity before the mid-Cenomanian and mid-Turonian events.

Trace element studies indicate that volcanic activity of the Caribbean Large Igneous Province increasingly supplied nutrients and sulphate to a low-sulphate ocean, with a major pulse ~500 kyr before OAE2 (Snow et al., 2005; Jenkyns et al., 2007; Turgeon and Creaser, 2008; Adams et al., 2010). This is manifested

Adams, D., Hurtgen, M., and Sageman, B.: Volcanic triggering of a biogeochemical cascade during Oceanic Anoxic Event 2, Nat. Geosci., 3, 201-204, doi:10.1038/ngeo743, 2010.

Berger, A., Loutre, M. F., and Laskar, J.: Stability of the Astronomical Frequencies Over the Earth's History for Paleoclimate Studies, 255, 560-566, 10.1126/science.255.5044.560, 1992.

Caron, M., Dall'Agnolo, S., Accarie, H., Barrera, E., Kauffman, E., Amédro, F., and Robaszynski, F.: High-resolution stratigraphy of the Cenomanian–Turonian boundary interval at Pueblo (USA) and wadi Bahloul (Tunisia): stable isotope and bio-events correlation, Geobios, 39, 171-200, doi:10.1016/j.geobios.2004.11.004, 2006.

Cleveland, W. S.: Robust Locally Weighted Regression and Smoothing Scatterplots, J. Amer. Statist. Assoc., 74, 829-836, doi:10.1080/01621459.1979.10481038, 1979.

Coccioni, R.: The Cretaceous of the Umbria-Marche Apennines (central Italy), Wiedmann Symposium bCretaceous Stratigraphy, Paleobiology and PaleobiogeographyQ, the Umbria–Marche Apennines (Central Italy). Tübingen, 1996, 7-10,

Du Vivier, A., Selby, D., Sageman, B., Jarvis, I., Gröcke, D., and Voigt, S.: Marine $^{187}Os/^{188}Os$ isotope stratigraphy reveals the interaction of volcanism and ocean circulation during

Oceanic Anoxic Event 2, Earth Planet. Sci. Lett., 389, 23-33, doi:10.1016/j.epsl.2013.12.024, 2014.

Gale, A., Kennedy, W., Voigt, S., and Walaszczyk, I.: Stratigraphy of the Upper Cenomanian–Lower Turonian Chalk succession at Eastbourne, Sussex, UK: ammonites, inoceramid bivalves and stable carbon isotopes, Cretaceous Res., 26, 460-487, doi:10.1016/j.cretres.2005.01.006, 2005.

[revised manuscript text omitted]